# Calcium promotes persistent soil organic matter by altering microbial transformation of plant litter

Itamar A. Shabtai [1,7] ✉, Roland C. Wilhelm [1,2], Steffen A. Schweizer [3], Carmen Höschen [3], Daniel H. Buckley [1,4] & Johannes Lehmann [1,5,6]

Calcium (Ca) can contribute to soil organic carbon (SOC) persistence by mediating physico-chemical interactions between organic compounds and minerals. Yet, Ca is also crucial for microbial adhesion, potentially affecting colonization of plant and mineral surfaces. The importance of Ca as a mediator of microbe-mineral-organic matter interactions and resulting SOC transformation has been largely overlooked. We incubated $^{44}$Ca labeled soils with $^{13}$C$^{15}$N labeled leaf litter to study how Ca affects microbial transformation of litter and formation of mineral associated organic matter. Here we show that Ca additions promote hyphae-forming bacteria, which often specialize in colonizing surfaces, and increase incorporation of litter into microbial biomass and carbon use efficiency by approximately 45% each. Ca additions reduce cumulative $CO_2$ production by 4%, while promoting associations between minerals and microbial byproducts of plant litter. These findings expand the role of Ca in SOC persistence from solely a driver of physico-chemical reactions to a mediator of coupled abiotic-biotic cycling of SOC.

Globally, there is a positive correlation between exchangeable calcium (Ca), mostly found on clay minerals, and soil organic carbon (SOC) content in slightly acidic to alkaline soils[1,2]. The conventional explanation for this relationship is that soil Ca can reduce SOC bioavailability and increase SOC stocks by driving physicochemical associations between organic compounds and minerals, such as sorption[3–5], co-precipitation and complexation[6,7], and occlusion within aggregates[8]. These mechanisms promote what is operationally defined as mineral-associated organic matter (MAOM), which has been shown to persist for longer timescales than bulk SOC[9–13]. While plant C is the principal source of SOC, plant C is to a large extent metabolized, and microbial products contribute significantly to MAOM[14]. MAOM formation is therefore often conceptualized as a two-step process: (1) microbial transformation and assimilation of plant C, and (2) the

association of microbial products, and soluble plant compounds, with soil minerals. The current paradigm for how Ca affects SOC persistence overlooks its potential influence over the first step, the microbial transformation of plant C.

The fate of plant and soil C is affected by its composition, C:N ratio, the microbial C use efficiency (CUE), and abiotic conditions (e.g., soil water content, mineralogy, and pH)[15–17]. The microbial processing of plant C can also be impacted by soil Ca contents[18,19], since Ca is a key factor in the growth and activity of fungi and bacteria, in particular surface-adhering[20–22] and biofilm-forming bacteria,[20–23] as well as fungal lignin-degrading enzymes[24]. Enrichment of bacterial taxa that attach to particulate C sources (i.e., plant litter or MAOM) may limit SOC gain due to enhanced litter mineralization[19], or conversely, may enhance SOC gain by limiting litter decomposition[18]. This trade-off can

[1]Soil and Crop Sciences, School of Integrative Plant Science, Cornell University, Ithaca, NY 14850, USA. [2]Department of Agronomy, College of Agriculture, Purdue University, West Lafayette, IN 47907, USA. [3]Chair of Soil Science, TUM School of Life Sciences, Technical University of Munich, 85354 Freising, Germany. [4]Department of Microbiology, Cornell University, Ithaca, NY 14850, USA. [5]Cornell Atkinson Center for Sustainability, Cornell University, Ithaca, NY 14850, USA. [6]Institute for Advanced Study, Technical University of Munich, Garching 85748, Germany. [7]Present address: Department of Environmental Science and Forestry, The Connecticut Agricultural Experiment Station, New Haven, CT 06511, USA. ✉e-mail: itamar.shabtai@ct.gov

be reflected in the quality and quantity of microbial biomass and metabolic by-products generated during litter decomposition, potentially impacting their downstream interactions with minerals, and SOC accrual[25].

Direct evidence for the effects of Ca on microbial SOC cycling are available from ecosystem-scale liming experiments performed in acidic, organic forest soils[18,26–28]. Such liming experiments produce concomitant changes in soil pH and Ca whose effects can be difficult to disentangle. Nonetheless, addition of wollastonite ($CaSiO_3$) or calcitic lime ($CaCO_3$) has been found to decrease SOC mineralization rates, and an associated long-term accumulation of C in organic soil horizons[18,27–29]. While some laboratory experiments have shown that elevated Ca has contrasting effects on litter or SOC mineralization[18,19,30], others have shown that lime and gypsum additions can alter the soil microbiome and increase SOC content[31,32]. Thus, despite evidence suggesting an underlying role of Ca in limiting decomposition[18,26,29], the specific roles of Ca in microbial transformation of plant C, and subsequent transfer of microbial products to MAOM, remain unclear.

Here, we tested the hypothesis that Ca promotes SOC persistence by altering microbiome structure and function in ways that govern plant litter decomposition, and by altering the formation of microbial products available for stabilization on mineral surfaces. Specifically, we asked (1) whether Ca treatment will select for microbial taxa that specialize in a surface-attached lifestyle, (2) what effects this shift has on CUE and microbial uptake of litter, (3) how Ca-induced changes to the microbiome affect the formation of MAOM, and (4) what the specific spatial and chemical roles are that Ca plays in these organo-mineral interactions. To address these questions, we used microcosm experiments to determine whether adding Ca to soil favors microbial transformation of plant litter into microbial biomass and MAOM (Fig. 1). We incubated a $^{44}CaCl_2$-labeled Mollic Endoaqualf silt-loam soil from a fallow agricultural field with $^{13}C^{15}N$-labeled willow leaf litter for ~4 months at the lower and higher ranges of water-filled pore space under non-drought conditions, which is a strong abiotic control on microbial activity. We measured total $CO_2$ and litter-derived $^{13}CO_2$ production throughout the incubation and sampled the microcosms at 4 days and ~4 months to determine the effects of elevated Ca on bacterial community and the transfer of leaf litter into microbial biomass and MAOM. FTIR-spectromicroscopy was used to study the formation of organo-mineral associations at the micron scale, while bulk near-edge X-ray absorption fine structure (NEXAFS) spectroscopy was used to evaluate the chemical speciation of C and Ca. Leveraging the $^{44}Ca$-labeling, we used nano-scale secondary ion mass spectrometry (NanoSIMS) to quantify the spatial co-localization of $^{44}Ca$, $^{15}N$, and minerals, in intact soil aggregates, to evaluate the effect of Ca on the formation of MAOM. Our results show a hitherto understudied mechanism of Ca-driven changes to the soil microbiome that alters the conversion of plant-derived C into more persistent MAOM fractions.

## Results

### Calcium alters microbial respiration, litter metabolism, and conversion to MAOM

We compared the effects of Ca on the metabolic processing of litter to test our hypothesis that Ca promotes SOC persistence. Calcium addition reduced cumulative $CO_2$ production at $T_{beg}$ (after 4 days; $P = 0.011$ and $P = 0.064$ under high and low water content, respectively) and at $T_{end}$ (after ~4 months; $P = 0.017$ and $P = 0.033$ under high and low water content, respectively) (Fig. 2a) by approximately 4% (Supplementary Table 2). However, Ca addition did not affect the proportion of $CO_2$ derived from litter mineralization at low or high-water content (Fig. 2b). Soils treated with KCl, instead of $CaCl_2$, and incubated with litter for 16 days had similar mineralization rates to control soils and greater mineralization than Ca-treated soils, demonstrating that the effects of the $CaCl_2$ treatment were due to Ca (Supplementary Fig. 1). We did not observe a significant interaction between water content and Ca addition on most of the studied variables (Supplementary Tables 2, 3, 4, and 5), and so we present the pairwise comparison between control and Ca-treated soils using the combined data (low- and high-water content). The full statistical analysis of each treatment factor is provided in the Supplementary information section (Supplementary Tables 2, 3, 4, and 5).

We also observed effects of Ca addition on microbial growth and metabolic efficiency. Ca addition increased microbial CUE at $T_{beg}$ ($P = 0.002$) (Fig. 2c) from 0.20 to 0.29 and increased the proportion of MBC from $^{13}C$-labeled plant litter at $T_{beg}$ ($P = 0.01$) (Fig. 2d) from 0.36 to 0.53, an approximately 45% increase for both measures. At $T_{end}$, the proportion of MBC from litter-C was still higher in Ca-treated soils ($P = 0.008$) while CUE was not different ($P = 0.08$) (Supplementary Table 2). Total MBC and dissolved organic carbon (DOC) were not affected by Ca addition at either $T_{beg}$ or $T_{end}$ (Supplementary Fig. 5).

The effects of Ca on litter decomposition impacted the transfer of litter C and N to the MAOM fraction. At $T_{beg}$, the proportion of litter-derived C and N in the MAOM fractions was, respectively, 16 and 14% lower in Ca-treated than control soils ($P = 0.014$ for C; $P = 0.043$ for N) (Fig. 2e), Supplementary Table 3). By $T_{end}$, a higher proportion of MAOM C and N was derived from litter in soils with high water content, but there was no difference in litter-derived MAOM content between treatments (Fig. 2e, Supplementary Table 3). However, at $T_{end}$, the C:N ratio of litter-derived $^{13}C^{15}N$ on the MAOM fraction in Ca-treated soil was 0.52 lower than control across both water contents ($P = 0.002$) (Fig. 2f, Supplementary Table 3). The amount of MAOM (mg g$^{-1}$ soil) (Supplementary Fig. 6c) or total C and N content in MAOM

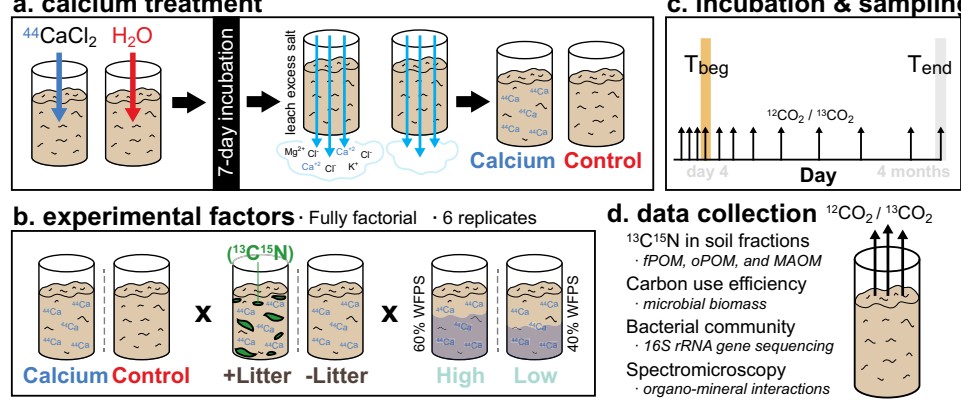

**Fig. 1 | Overview of experimental methods, design, and data collection.** In (**a**), our method for pre-incubating soil with $^{44}CaCl_2$ to elevate soil calcium. In (**b**), a list of all experimental factors. In (**c**), a schematic illustrating the incubation length and sampling timepoints. In (**d**), a list of data types collected at $T_{beg}$ (4 days) and $T_{end}$ (~4 months).

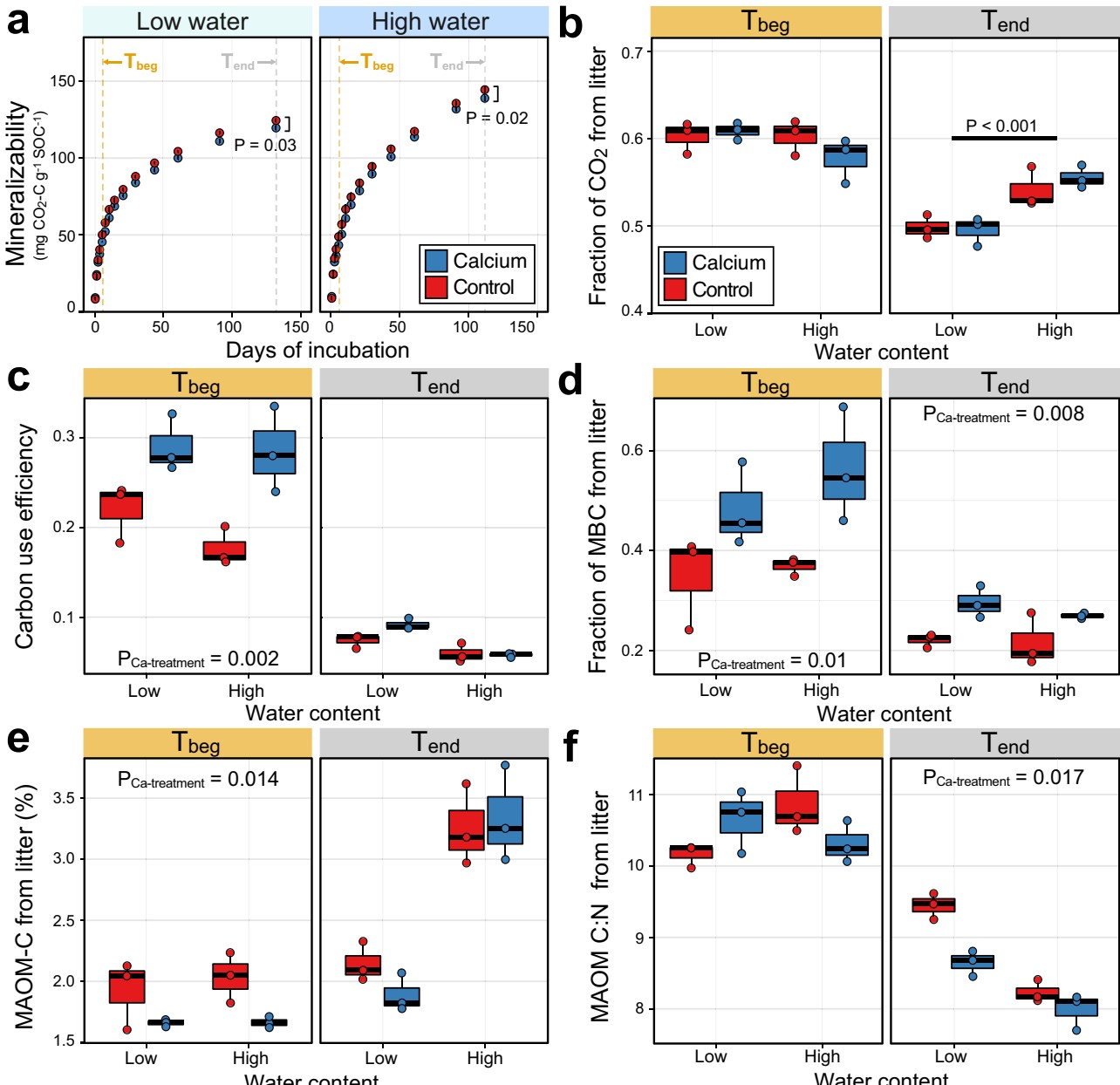

**Fig. 2 | Calcium reduces soil heterotrophic respiration by increasing assimilation of litter-$^{13}$C into microbial biomass and inducing transfer of highly processed litter into mineral-associated organic matter (MAOM) fractions.** Cumulative mineralizability (**a**) and fraction of $CO_2$ derived from $^{13}$C$^{15}$N enriched litter (**b**) in Ca-treated and control soils incubated at low or high water content and sampled at $T_{beg}$ and $T_{end}$. The cumulative mineralizability was significantly greater at high water content at $T_{end}$ ($P < 0.001$), averaged across treatment, but did not differ by water content at $T_{beg}$. Microbial carbon use efficiency (CUE) (**c**), fraction of microbial biomass C (MBC) derived from $^{13}$C$^{15}$N labeled litter (**d**), percentage of

MAOM-C from litter-$^{13}$C (**e**), and the C:N ratio of litter-derived $^{13}$C$^{15}$N on the MAOM fraction (**f**) in Ca-treated and control soils incubated at low or high water content and sampled at $T_{beg}$ and $T_{end}$. All statistics were derived from $n = 3$ independent samples. Error bars in panel A denote the standard deviation from the mean. Box plot center line represents median, box limits the first and third quartiles, whiskers extend to the smallest and largest values no further than 1.5 × interquartile range (IQR). The sample size '$n$' represents independent samples. Statistical significance was tested using two-sided, unpaired $t$-tests.

(Supplementary Fig. 7a, c) did not differ between treatments. The amount of occluded particulate organic matter (oPOM) averaged across both water contents was slightly higher in control than Ca-treated soils at $T_{end}$ ($P = 0.04$) (Supplementary Table 4, Supplementary Fig. 6b), but, overall, Ca treatment did not significantly affect the proportion of litter-$^{13}$C or $^{15}$N in free particulate organic matter (fPOM) and oPOM fractions (Supplementary Table 5). This finding suggests that the occlusion of litter within aggregates was not an important mechanism by which Ca reduced respiration or altered microbial litter metabolism in our experiment.

## Calcium-induced shifts in bacterial community composition

Bacterial community composition was significantly impacted by all treatment factors at both time points, as evidenced in the clustering of samples in a t-SNE ordination (Fig. 3a). At $T_{Beg}$, Ca and litter addition explained 28% and 32% of the variation among bacterial communities, respectively. At $T_{end}$, Ca addition explained an increasing proportion of variation in bacterial community composition (32%), more than explained by litter amendment (18%) (Fig. 3b). Moisture had a significant effect on bacterial community composition, but it explained less variation than Ca or litter (Fig. 3b). Ca addition brought about an

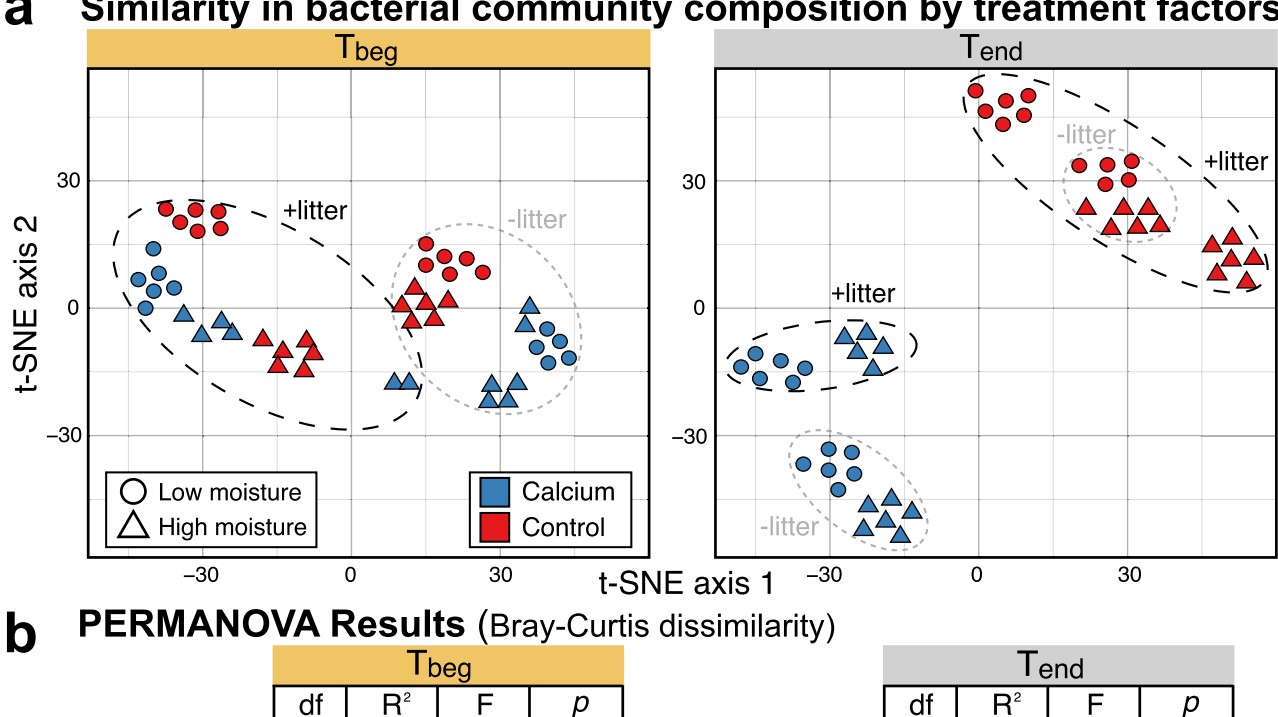

**a** Similarity in bacterial community composition by treatment factors

**b** PERMANOVA Results (Bray-Curtis dissimilarity)

| | Tbeg | | | | |
|---|---|---|---|---|---|
| | df | R² | F | p | |
| Litter | 1 | 0.32 | 41.6 | 0.001 | *** |
| Calcium | 1 | 0.28 | 36.3 | 0.001 | *** |
| Moisture | 1 | 0.07 | 9.1 | 0.001 | *** |
| *residual* | 43 | 0.33 | | | |
| *total* | 46 | 1 | | | |

| | Tend | | | | |
|---|---|---|---|---|---|
| | df | R² | F | p | |
| Litter | 1 | 0.18 | 19.7 | 0.001 | *** |
| Calcium | 1 | 0.32 | 34.2 | 0.001 | *** |
| Moisture | 1 | 0.1 | 11.0 | 0.001 | *** |
| *residual* | 42 | 0.39 | | | |
| *total* | 45 | 1 | | | |

**c** Ca-induced shifts in litter-associated populations

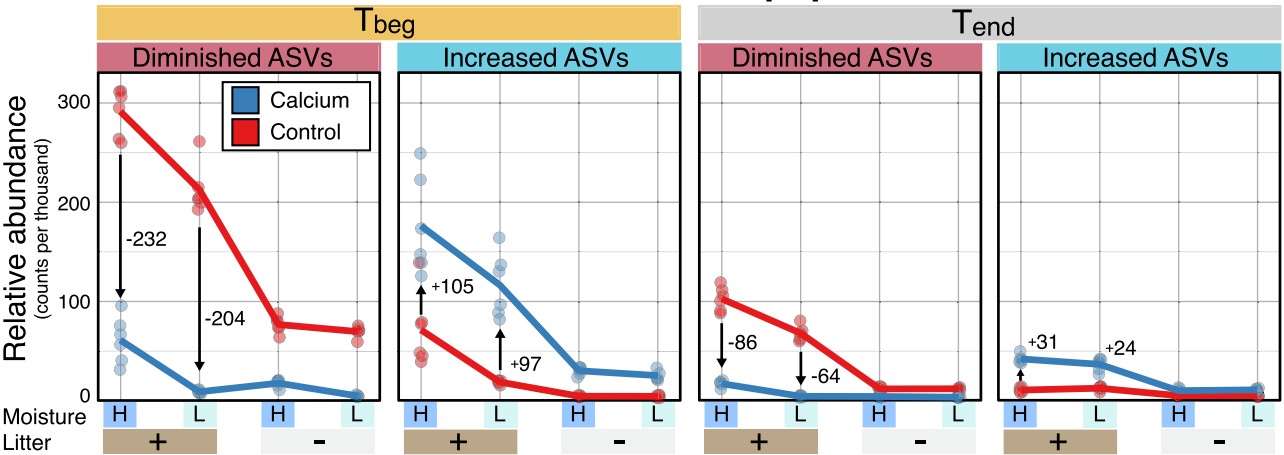

**Fig. 3 | Ca-induced shifts in bacterial community composition were pronounced and strengthened over time, and disproportionately resulted in a relative decrease in litter-associated populations.** In (**a**), a t-SNE ordination illustrates the divergence of bacterial communities according to treatment factors according to Bray-Curtis dissimilarity. In (**b**), variation in community composition was most attributed to litter amendment at $T_{beg}$ according to a PERMANOVA test of treatment factors based on Bray-Curtis dissimilarity. By $T_{end}$, variation in composition was most attributed to Ca treatment (see **b**), which was also apparent in the separation along both principal ordination axes (see A). In (**c**), plotting the aggregated relative abundance of amplicon sequence variants (ASVs) that were differentially more abundant in litter-amended soils (i.e., putative decomposers) revealed a disproportionate decrease in Ca-treated soils. Arrows indicate the average increase or decrease in relative abundance. Note the greater size of arrows for the 'diminished ASVs' relative to 'increased ASVs' for each timepoint. All statistics were derived from $n = 6$ independent samples. Statistical significance was tested using a PERMANOVA test (number of permutations = 999).

immediate and enduring enrichment of many bacteria that belong to the phyla *Bacillota* (class *Bacilli*) and *Actinomycetota* (classes *Actinomycetes* and *Thermoleophilia*) as well as select members of *Pseudomonadota* (Supplementary Fig. 10). While most *Pseudomonadota* declined in relative abundance in response to Ca addition, especially members of the *Gammaproteobacteria* (orders *Burkholderiales* and

*Pseudomonadales*) and *Alphaproteobacteria* (*Caulobacterales*), a few major groups were favored by Ca (orders *Hyphomicrobiales* and *Xanthomonadales*) and these populations had increased relative abundance compared to controls only at $T_{end}$ (Supplementary Fig. 11). Notably, Ca caused a decline in relative abundance of bacterial populations that were enriched in response to litter (see arrows in Fig. 3c),

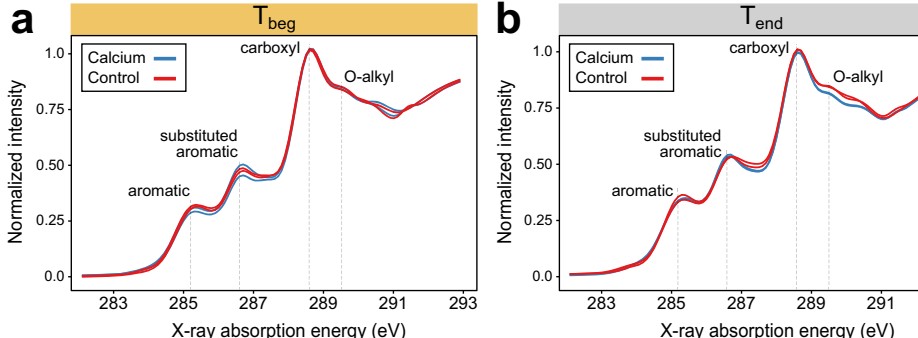

**Fig. 4 | Plant litter transformation was enhanced by calcium addition.** Bulk soil SOC chemical composition using C K-edge NEXAFS of Ca-treated and control soils incubated with litter and sampled at $T_{beg}$ (**a**) or $T_{end}$ (**b**). Samples incubated at low and high water contents are shown for each treatment.

indicating an effect of Ca on putative decomposers populations. A complete list of ASVs that were differentially abundant according to treatment factors is available in Supplementary Dataset 1, as well as further characterizations of the changes in the soil microbiome are provided in the Supplementary Information and Supplementary Figs. 8–11.

### Ca influences SOC chemical composition and spatial distribution

In addition to the effects of Ca on microbial community structure and utilization of litter, Ca addition resulted in changes to bulk SOC composition, as measured by C K-edge NEXAFS spectroscopy. At $T_{beg}$, SOC composition was similar across treatments, but by $T_{end}$ (after >40% of the litter had been mineralized), Ca-treated soils had lower relative abundance in O-alkyl C content (289.4 eV) than controls, indicating greater decomposition of the carbohydrate component of litter in Ca treated soil (Fig. 4, Supplementary Table 6). This is consistent with the results that show enhanced incorporation of litter into microbial biomass (Fig. 2d).

We also found, using infrared spectromicroscopy, that Ca affected the formation of organo-mineral associations (Fig. 5). The spatial co-localization of clay minerals (2:1 and 1:1 aluminosilicates such as smectite and kaolinite, as indicated by the absorbance of structural OH groups at 3620 cm⁻¹) and three organic functional groups (aromatic, aliphatic, and carboxylic) across intact microaggregate sections were higher in the Ca treated soil, as indicated by the higher coefficients of determination ($R^2$ values) both at $T_{beg}$ and at $T_{end}$ (Fig. 5b–d, 5f–h). Microaggregates from Ca-treated soils also had greater organic loading per unit mineral content, as indicated by significantly higher regression coefficients (Supplementary Table 7), and greater clay mineral abundances (greater absorbance at 3620 cm⁻¹). Lastly, we found that both $R^2$ values and coefficients in regression models for carboxylic-C and aromatic-C in microaggregates from Ca-treated soils, increased from $T_{beg}$ to $T_{end}$, while these values decreased for aliphatic-C (Fig. 5, Supplementary Table 8). In contrast, the greatest increase in $R^2$ and regression coefficient values in regression models of controls soils was in the relationship between clay minerals and aliphatic C (Fig. 5, Supplementary Table 8). These findings reflect the Ca-driven formation of organo-mineral associations with microbial transformation products and possibly soluble plant-derived compounds rich in carboxylic and aromatic functional groups.

We further investigated the role of Ca in MAOM formation at $T_{end}$ by tracking ⁴⁴CaCl₂-derived ⁴⁴Ca and litter-derived ¹⁵N on mineral surfaces using NanoSIMS (Fig. 6). NanoSIMS images reveal a heterogeneous arrangement of minerals, pores, and organic matter within sections of intact aggregates (Supplementary Fig. 14). Images were segmented into mineral-dominated and organic matter-dominated regions (Fig. 6a) and each region was further classified as dominated

by Ca, aluminum (Al), or iron (Fe) (Fig. 6b). Widespread ⁴⁴Ca enrichment was observed across both mineral- and organic matter-dominated regions (Supplementary Fig. 14d), averaging 49.2 atom % in organic matter regions and 42.6 atom % in mineral regions (Supplementary Fig. 15a and 15f). Ca addition increased co-localization of Ca with mineral regions from 11% to 14% (Fig. 6d) and increased Ca co-localization with organic matter regions from 68% to 90% (Fig. 6e). The effect of Ca addition on Ca co-localization with the organic matter was greater when Ca and litter were added together compared to when only Ca was added (Supplementary Fig. 15b). Enrichment in ¹⁵N and normalized N:C were higher in Ca-dominated segments of mineral regions of Ca-treated soils compared to control soils (Fig. 6f, g, Supplementary Fig. 15h), which indicates that Ca addition enhanced mineral association with N-rich organic compounds. The contribution of inorganic-N species to this result is negligible since they made up, at most, ~1% of total N (Supplementary Fig. 16), and in addition, the fragment used for N detection (CN⁻) is mostly contributed to by organic N compounds[33]. These micron-scale observations at $T_{end}$ were consistent with bulk measurements which showed lower C:N from litter-derived ¹⁵N and ¹³C on the MAOM fraction of Ca-treated soils (Fig. 2d). Furthermore, when Ca was added, the ratio of Ca:N at the mineral surface doubled compared to control (Fig. 6h), highlighting the interactions of mineral-bound Ca with N-rich organic matter, some of which was co-localized with litter-derived ¹⁵N (Fig. 6c). These results show, through evident co-localization, that Ca interacted at the mineral-litter interface, especially when Ca and litter were applied together, and enhanced deposition of litter transformation products onto mineral surfaces, forming potentially new organic-mineral associations.

## Discussion

Current conceptual frameworks assign the influence of Ca on SOC persistence solely to physicochemical mechanisms such as sorption, co-precipitation, and occlusion of organic matter[2,6,34]. Our findings show that Ca addition exerted an immediate and prolonged influence on microbial community structure and litter metabolism, revealing upstream Ca effects on the biological properties of SOC cycling. We observed that Ca addition decreased soil respiration and increased the efficiency of litter C assimilation, resulting in the transfer of more highly processed litter C and N into the MAOM fraction. Our results support the hypotheses that Ca affects microbial community structure, altering microbial metabolism of litter, and promoting enhanced stabilization of these products through Ca-driven organo-mineral interactions. Here, we discuss the potential mechanisms underlying the changes in litter cycling caused by Ca additions.

We observed two major trends in the effects of Ca on the bacterial community. First, Ca had a disproportionately large negative effect on ASVs favored by litter addition (Fig. 3c). This observation suggests that

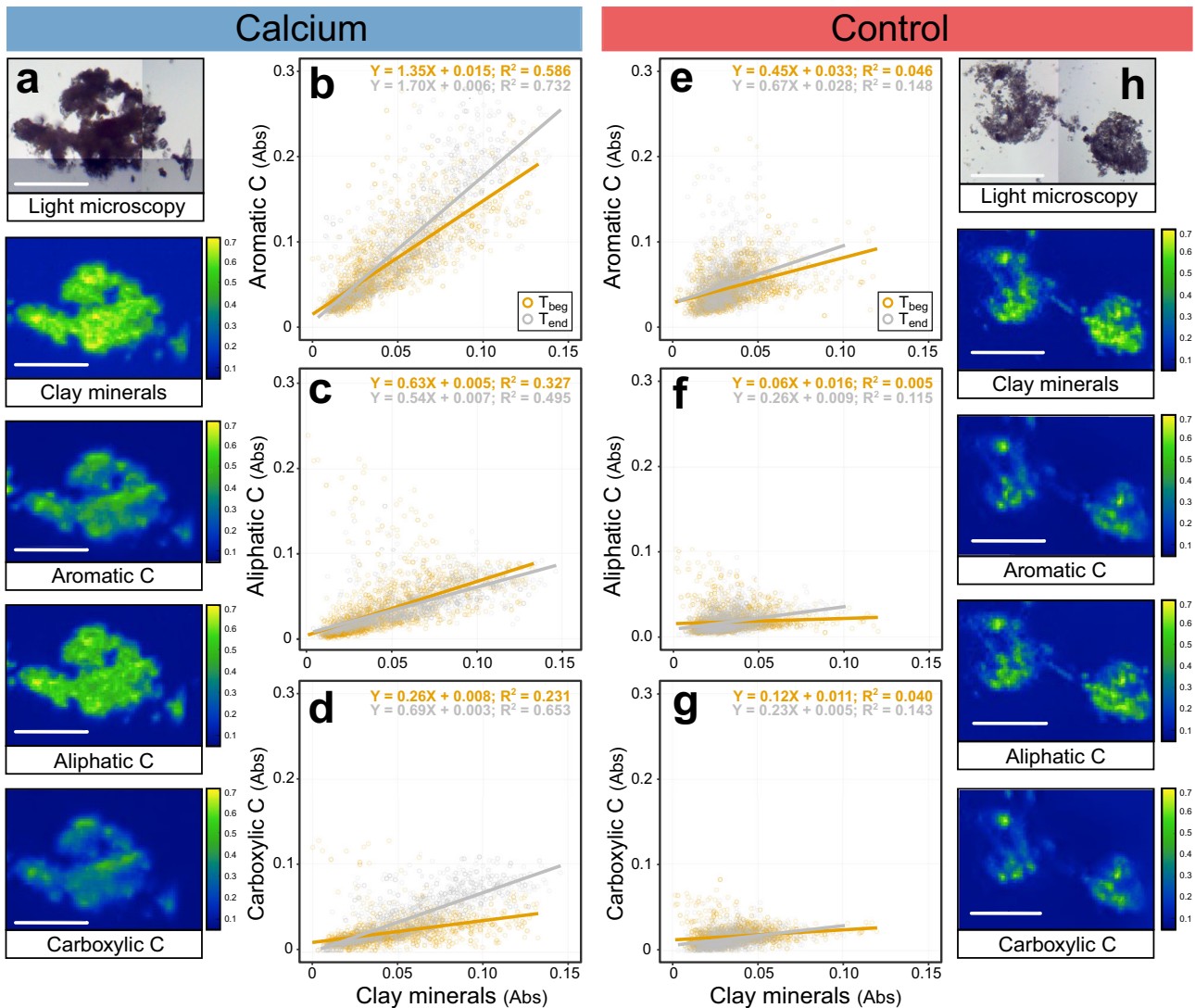

**Fig. 5 | FTIR spectromicroscopy analysis showing enhanced formation of organo-mineral associations between clay minerals and organic compounds in calcium-treated soils.** Exemplary optical microscopy and spectral maps showing the distribution of clay minerals (absorbance at 3620 cm$^{-1}$), aliphatic C (absorbance at 2920 cm$^{-1}$), aromatic C (absorbance at 1620 cm$^{-1}$), and carboxylic C (absorbance at 1413 cm$^{-1}$) across microaggregate sections taken from Ca-treated soils (**a**) and control soils (**e**) (white bars: 100 μm). The color scale corresponds with raw absorbance intensity for each molecular group. Regression analyses (summarizing results from $n = 2000$ data points collected from at least $n = 8$ sections for each sample) of the amount of aliphatic C (absorbance at 2920 cm$^{-1}$), aromatic C (absorbance at 1620 cm$^{-1}$), and carboxylic C (absorbance at 1413 cm$^{-1}$) as a function of the amount of clay minerals (absorbance at 3620 cm$^{-1}$) in microaggregate sections at $T_{beg}$ (orange circles) and $T_{end}$ (grey circles) of incubation at low water content (data from high water content soils not collected).

Ca treatment restructured decomposer communities, and that the reduced net respiration and increased CUE resulted to some extent from a change in decomposer populations. However, we cannot test this relationship directly, as we lack information about changes in the absolute abundance and activity of these populations. Second, Ca treatment enhanced the relative abundance of surface-colonizing bacterial taxa, including *Actinomycetota*, that form hyphae such as *Nocardioides*, *Rhodococcus*, *Leifsonia*, *Streptomyces*, *Cellulomonas*, *Agromyces*, and *Pseudonocardia*[35–38], and surface motile or surface-adhering populations such as *Devosia*, *Hyphomicrobium*, *Haliangium*, *Sorangium*, members of *Fibrobacterota*, *Asticcacaulis*, *Luteimonas*, and *Adhaeribacter*[39–46] (Supplementary Figs. 7, 8). Many of the same taxa were also similarly favored in a study of forest soil amended with calcitic lime, especially *Nocardioides*, *Devosia*, *Haliangium*, and *Hyphomicrobium*[27], suggesting that the observed shift in surface-colonizing bacteria is associated with Ca-driven processes. Characterization of changes in the fungal community composition may

yield additional insights given Ca is an important constituent of fungal cell walls and can affect the growth of fungal decomposers in forest soils[19]. However, bacteria tend to be more dominant than fungi in the decomposition of litter in the mineral soil (compared to surface litter decomposition), and in non-acidic soils[47], which was the focus of this study.

The abiotic and biotic factors driving changes in bacterial community composition, and the prevalence of surface-colonizing populations, cannot be disentangled using phylogenetic gene marker data, and are likely a result of a combination of factors (see Supplementary Information for more details). One likely contributing factor is the direct effect of Ca, which is often a critical co-factor in bacterial attachment and biofilm production[20–23], promoting the growth and activity of surface-adhering bacterial populations. The reduction in mineralization in Ca-treated soils likely reflect physiological and metabolic differences in surface-attached populations compared to bacteria adapted for growth in pore water, including slower growth

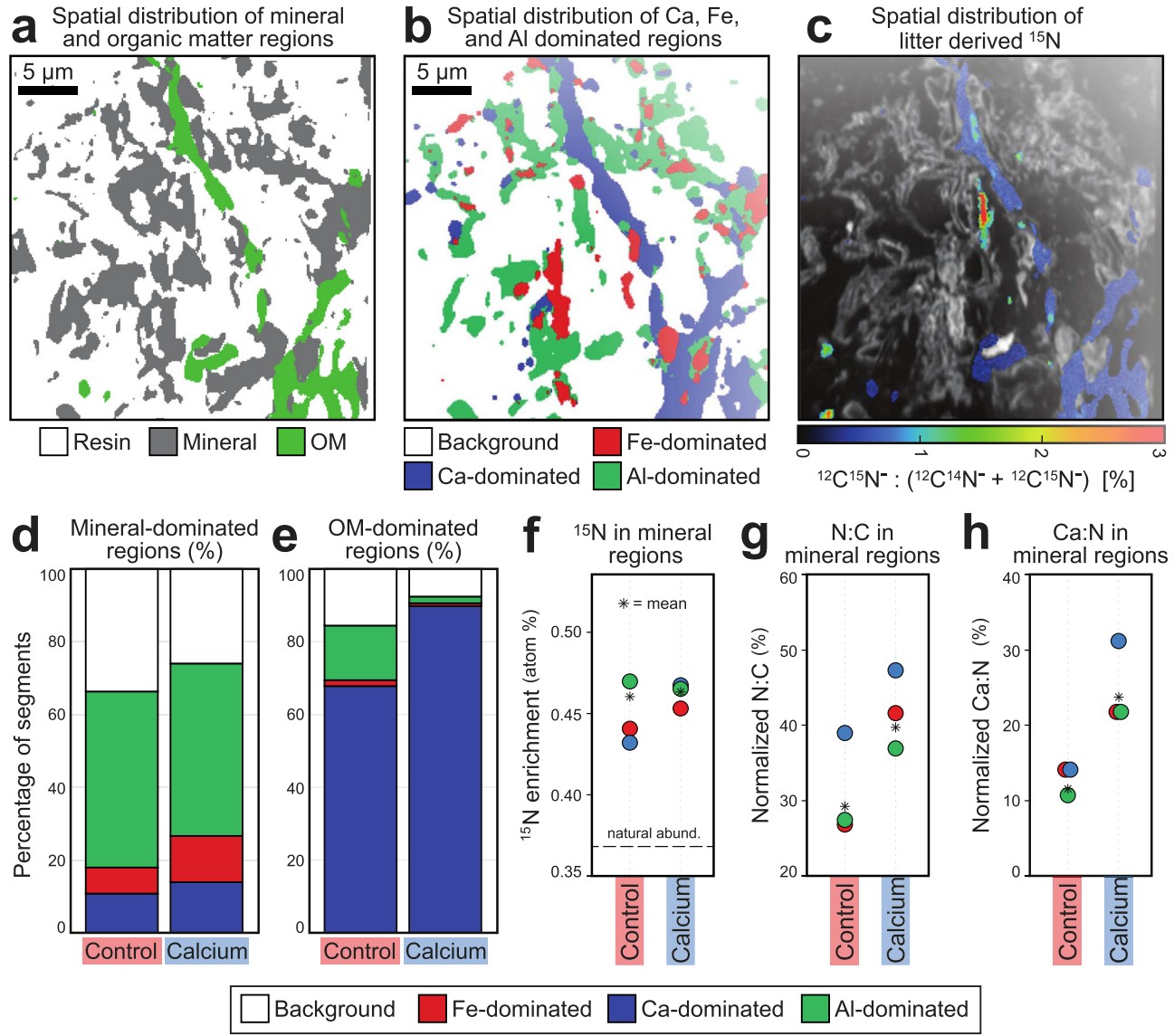

**Fig. 6 | NanoSIMS analysis showing that calcium mediates deposition of litter transformation products onto mineral surfaces.** Exemplary image of $^{44}$Ca-treated soil amended with $^{13}$C$^{15}$N litter and incubated at high water content. Regions dominated by resin, mineral and organic matter (OM) (**a**), spatial distribution of regions dominated by iron (Fe), calcium (Ca), aluminum (Al), or none of these, based on machine-learning segmentation (**b**), and spatial distribution of $^{15}$N enrichment (**c**). Area proportion of Ca, Al, and Fe co-localization with mineral dominated regions (**d**) and organic matter (OM)-dominated regions (**e**). Quantification of the $^{15}$N enrichment ($^{12}$C$^{15}$N$^-$:($^{12}$C$^{14}$N$^-$ + $^{12}$C$^{15}$N$^-$)) (**f**), normalized N:C ratio ($^{12}$C$^{14}$N$^-$):($^{12}$C$^-$ + $^{12}$C$^{14}$N$^-$) (**g**), and the Ca:N ratio (($^{40}$Ca$^+$ + $^{44}$Ca$^+$):($^{40}$Ca$^+$ + $^{44}$Ca$^+$ + $^{12}$C$^{14}$N$^-$ + $^{12}$C$^{15}$N$^-$)) (**h**). Bar and scatter plots show combined data from soils incubated at low and high water contents. See Supplementary Fig. 15 for separate results from low and high water content. In total, $n = 9$ images from control soils and $n = 12$ images from calcium-treated soils were analyzed.

rates, as well as increased association between microbial byproducts (e.g., adhesive proteins and extracellular polymeric substances) and mineral surfaces[48].

The Ca-induced shift in microbial processing of litter had a cascading effect on the C and N occurring as MAOM. In the Ca-treated soils, a larger fraction of MBC at $T_{beg}$ was derived from litter-C than in control soils, indicating that more litter was cycled through microbial biomass, while concomitantly less MAOM from litter was measured in Ca-treated soils. This is likely an observation of a transient nature at $T_{beg}$ which captures the greater uptake of litter-C into MBC prior to its conversion into MAOM. At $T_{end}$, we found that the amount of litter-derived C and N transferred to MAOM was not different (Fig. 2e), which may be explained by the fact that surface-adhering bacteria likely also consumed litter-derived MAOM throughout the incubation. The observation that the proportion of MBC derived from litter-C was significantly higher after Ca addition even at $T_{end}$ (Fig. 2e and

Supplementary Table 2) supports this claim. So does the observation at $T_{end}$ that MAOM consisted of compounds that were more microbially processed (lower C:N) after Ca addition (Fig. 2f). Thus, MAOM composition, but not amount, was impacted by Ca addition. Additional evidence of compositional changes in SOC upon Ca addition comes from bulk C NEXAFS analysis which shows lower relative O-alkyl abundance (289.4 eV) at $T_{end}$ than in the control soils (Fig. 4b). While this could have theoretically resulted from a shift in the composition of compounds produced by microbes after Ca addition, the most reasonable explanation is that litter decomposition was driving O-alkyl reduction, as previously reported for incubation studies[49].

We also found that Ca plays a direct role in the transfer of litter-derived decomposition products onto mineral surfaces – forming new organo-mineral associations. When Ca was added, we observed higher co-localization between clay minerals and organic compounds, greater organic loading, and increased clay mineral particle aggregation at

sub-micron scales (Fig. 5). This is consistent with other studies that have shown that Ca can play an important role in mediating organo-mineral interactions[5,7,50,51]. Furthermore, we found that the amount and co-localization of carboxylic-C and aromatic-C with clay minerals in Ca-treated soils increased with incubation time (Fig. 5 and Supplementary Table 8). This result indicates that organic matter accumulated in spatially constrained hotspots, as suggested by recent studies[52,53]. Such accumulation patterns may reflect the influence of Ca on the activity of microbial surface colonies which produce and deposit metabolic byproducts on mineral surfaces. A complementary explanation to our FTIR results is that Ca also promoted the sorption of soluble plant-derived compounds rich in aromatic and carboxylic moieties[13]. These observations are consistent with NanoSIMS measurements which showed increased co-localization of Ca and litter-derived $^{15}N$ on mineral surfaces (Fig. 6f and Supplementary Fig. 15h).

The discrepancy between results obtained with FTIR-microscopy and NanoSIMS analyses, which indicated greater abundance of functional groups in Ca-treated soils and co-localization with minerals, and elemental and isotope analysis that did not indicate such differences (Fig. 2e, Supplementary Fig. 7), could stem from the spatially resolved nature of the former vs bulk characterization of the latter, and from the fact that FTIR-microscopy was done on microaggregate (53-250 μm) sections while the isotope analysis was done on <53 μm particles, which may have accrued different amounts of C. Additionally, we cannot rule out the possibility that the sodium polytungstate and sodium hexametaphosphate used in floatation and dispersion during the fractionation protocol may have exchanged Ca ions, potentially destabilizing some of the organic matter that was bound to the added Ca[54,55]. While Na-polytungstate is currently the best option for density fractionation, a less harsh procedure for dispersal (e.g., shaking with glass beads) may be called for.

Reduced C bioavailability due to Ca-driven sorption or precipitation of DOC did not appear to be the direct cause of lower mineralization rates, since DOC concentrations were comparable, or slightly higher, in Ca-treated than control soils (Supplementary Fig. 5). Had DOC concentration been a driver of mineralization rates, the opposite observation would have been expected. Our observations contradict other studies that found reduced DOC concentrations following a Ca treatment[18,30], however, the soils in those studies had substantially lower pH, higher SOC, and lower exchangeable cation capacity, which complicate a comparison with our findings. Preliminary observations

also ruled out the effects of transient increase in soil salinity on mineralization with a KCl control (Supplementary Fig. 1), which showed that when soils were treated with an equivalent concentration of KCl, the resulting mineralization over 16 days was similar to control and higher than the Ca-treated soil. This finding supports our conclusion that Ca was responsible for the changes to mineralization, bacterial community composition, and C metabolism. Furthermore, if lower DOC bioavailability or saline conditions were limiting factors, litter transformation in Ca-treated soils would also be reduced and CUE would have likely decreased[56,57], which was not the case (Fig. 2d, Fig. 4). Further discussion on the possible role of saline conditions during preincubation is provided in the Supplementary Information.

Taken together, our results support our hypotheses that Ca additions affect microbial community structure, altering microbial metabolism of litter, and promoting enhanced stabilization of microbial byproducts through Ca-driven organo-mineral interactions. The Ca addition increased an already high exchangeable-Ca content (from 85% to 92% of the soil's cation exchange capacity - CEC) (Supplementary Table 1) but did not change Ca speciation (Supplementary Figs. 2, 3), which raises the question: how did a minor increase in exchangeable Ca bring about a major change in both microbial processing and MAOM formation?

Based on our findings we conclude that the most likely mechanism (Fig. 7) was that $Ca^{2+}$ added to the soil solution as $CaCl_2$ displaced other exchangeable cations, mostly $Mg^{2+}$ and $Ca^{2+}$, providing pristine sites for microbial attachment and adsorption of litter transformation products. Additionally, the elevated concentration of soluble Ca ions likely caused changes in microbial cell regulation which triggered a switch to surface colonization, as previously reported[58–61]. Despite both $Mg^{2+}$ and $Ca^{2+}$ being divalent cations, they likely function differently. Indeed, organic matter affinity to $Ca^{2+}$ was found to be greater than to $Mg^{2+}$ (ref. 62, 63). $Ca^{2+}$ is also required for inducing surface-colonizing activity, such gliding motility, and cannot be replaced by $Mg^{2+}$ (ref. 21). Subsequently, surface-attached microbial cells, microbial by-products and necromass may have been deposited at greater loadings on mineral surfaces[64,65]. Ca-driven organo-mineral associations of highly processed organic molecules likely reduced microbial access, increasing their persistence in soil. The latter explanation indicates that the biotic effect of Ca on microbial C cycling preceded the abiotic effect. Regardless, both scenarios suggest that even small changes in the concentration of soluble Ca can influence SOC cycling and persistence, whether through cation

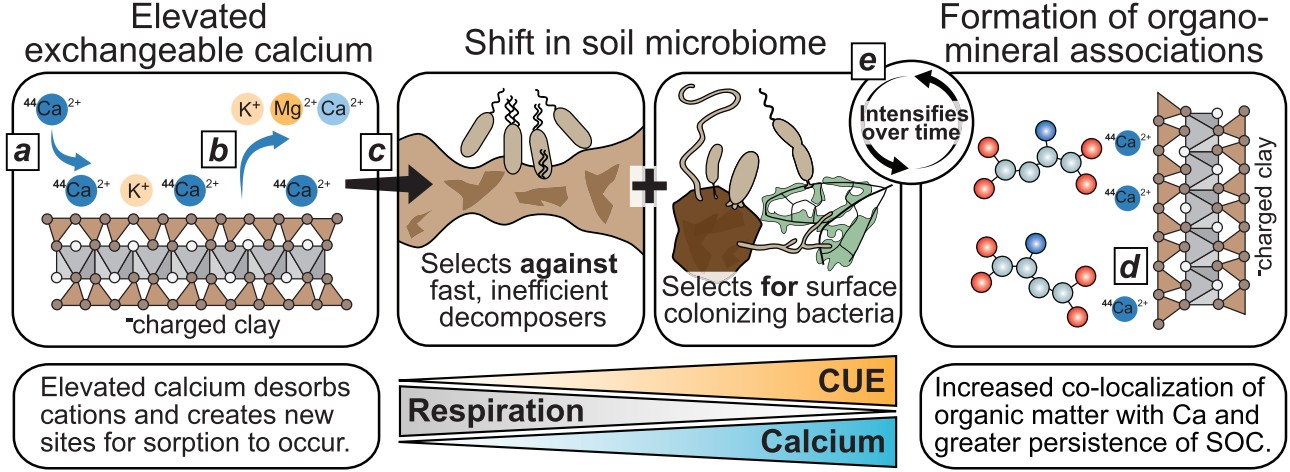

**Fig. 7 | Calcium promotes mineral-associated soil organic matter by mediating coupled biotic-abiotic mechanisms.** $^{44}$Calcium additions to soil increased exchangeable $^{44}$Ca content (**a**) and desorbed some native cations (**b**). Elevated calcium concentration promoted efficient surface colonizing bacteria and selected against inefficient decomposers (**c**), increasing carbon use efficiency, and decreasing microbial respiration. Calcium enhanced the formation of organo-mineral association with microbial byproducts of plant litter (**d**). Localization of organic compounds on surfaces reinforced selection for surface colonizing bacteria and deposition of microbial products on mineral surfaces, intensifying this cycle with time (**e**).

exchange processes on mineral surfaces or biochemical regulation of surface-attaching behavior.

Our study revealed the influence of Ca on previously overlooked biological mechanisms that affect MAOM formation. While the extent that Ca drives biotic controls over SOC cycling will be soil dependent, our findings suggest that amendments, which raise soluble Ca content can impact the conversion of organic inputs to MAOM, a more persistent form of SOC. These observations raise the possibility that Ca additions may be used to manage SOC in agroecosystems. For example, Ca-containing soil amendments, such as gypsum and lime, are commonly added to manage soil fertility, while basalt rock (calcium silicate) is applied to sequester inorganic C through enhanced rock weathering. These amendments may concomitantly impact SOC due to effects on microbial C cycling, especially when coupled with organic matter amendments. Understanding the nature of the soil Ca pools responsible for coupling biotic-abiotic SOC cycling processes will help optimize the management of Ca amendments and SOC stocks for more climate-smart soils.

## Methods

### Soil sampling

A Mollic Endoaqualf silt loam was collected at 0 – 15 cm from a fallow subplot located in an experimental bioenergy feedstock field trial in Ithaca, NY (42° N 28.20', 76° W 25.94'). The field has not been plowed since at least 1954, and since then has been under perennial grass and broadleaf forbs cover and has been mowed every several years. The field has been completely fallow since 2005 and has received no fertilizer. The mean annual temperature and precipitation at the site is 10 °C and 940 mm, respectively. Soil was air-dried, sieved to 2 mm, and visible plant detritus removed. The soil texture, measured with the hydrometer method after dispersion with sodium hexametaphosphate, is 22.5% sand, 57.5% silt, and 20.0% clay. Soil total C and N contents are 3.66% and 0.36%, and $\delta^{13}C$ and $\delta^{15}N$ values are -27.6 ‰ and 5.3‰, respectively. Soil pH and electrical conductivity are 6.1 and 184 µS cm$^{-1}$, respectively, in a 1:2.5 soil:water extract (Supplementary Table 1). Exchangeable cations were extracted using 1 N ammonium acetate at pH 7 and measured on an inductively coupled plasma spectrometer (iCAP 6000 series, Thermo Fisher Scientific, Waltham, MA, USA). Soil maximum water-filled pore space (WFPS) was calculated as the weight difference after saturating 10 g of air dry soil (packed in a 5 cm diameter plastic tube) with water overnight, and allowing it to drain for one hour.

### Experimental design

For reference, an overview of the three-factor factorial experimental design is provided in Fig. 7. A $CaCl_2$ solution at a concentration of 200 meq L$^{-1}$ (Ca treatment) or deionized water (control treatment) were added to the soil to reach a water content of 60%WFPS. Soils were then pre-incubated for one week at 20°C in Mason jars, after which all soils were leached on a filter funnel fitted with filter paper with consecutive 50 mL portions of deionized water until leachate electric conductivity was <100 µS cm$^{-1}$, then soils were allowed to air dry. This method was chosen to increase soil Ca without concomitantly changing soil pH. Our approach was chosen over shaking a soil slurry with a $CaCl_2$ solution[26] to minimize the impact on soil structure. This approach also allowed us to leach excess salt at the end of this phase. The effect of salinity during Ca treatment on subsequent C dynamics was tested in a preliminary experiment in which soils were treated with KCl (200 meq L$^{-1}$) as above, incubated with plant litter for 16 days, and heterotrophic respiration was measured (Supplementary Fig. 1). Treatment with $CaCl_2$ resulted in a ~10% increase in exchangeable Ca, but did not change soil pH, EC, or aggregate slaking value compared to control (Supplementary Table 1). Dissolved organic carbon (DOC) in the Ca-treated soil was higher than in the control (Supplementary Table 1), which may have been due to the release into the solution of C associated with native Ca that was exchanged by added Ca. Based on

NEXAFS spectroscopy of the soil samples, Ca treatment did not alter Ca L-edge and there was no evidence of excess $CaCl_2$ (Supplementary Figs. 2, 3, 4) (See Supplementary Information for details).

Soil microcosms consisted of air-dry soil (9 g, <2 mm) mixed with labeled plant litter (2% by weight, <1 mm) in 60-mL Qorpak vials. Litter was $^{13}C^{15}N$-labeled (1.895 atom% $^{13}C$; 8.203 atom% $^{15}N$) willow leaves (*Salix viminalis x S.miyabeana*)[66]. Ca content in the willow leaves was ~15 mg g$^{-1}$ dry weight, as determined using acid digestion. Litter addition to soil corresponded to 1.56 ± 0.03 meq Ca 100 g$^{-1}$ soil, or ~10% of initial exchangeable Ca after Ca treatment. Samples were wetted with deionized water and incubated (20°C) in a 2-by-2 factorial design with pre-incubation (Ca vs control) and water content, 'low' (40% WFPS) or 'high' (60% WFPS) as treatments. Soil microcosms without litter were included for each treatment combination. Moisture was included as a factor since it strongly influences both substrate availability and biotic activity[67,68].

Six microcosms were prepared for each treatment and the whole sample set was duplicated and destructively sampled after four days ($T_{beg}$) and at the end of the incubation ($T_{end}$), after 112 days (for high water content) or 135 days (for low water content). For NanoSIMS analysis (see below), a third set of microcosms was incubated with 1 g of soil treated with $^{44}CaCl_2$ solution (97 atom%, Sigma Aldrich), and $^{13}C^{15}N$ labelled litter (2% w/w) at low or high water content and sampled only at $T_{end}$. Headspace $CO_2$ concentrations on the first two sets of microcosms were measured on three of the six microcosms prepared for each treatment combination using KOH traps on days 1, 2, 3, 4, 6, 8, 11, 15, 21, 30, 44, 61, 91, and $T_{end}$. Integrated measurement of $^{13}CO_2$ in days 0-4 ($T_{beg}$) and 5-112 (high WFPS) or 5-135 (low WFPS) ($T_{end}$) was obtained using the $Ba^{13}CO_3$ method[69]. Further details on respiration measurements are provided in the Supplementary Information.

We ended the incubation after a calculated 41.0% and 47.5% of the litter was mineralized under low or high water content, respectively. To estimate litter mineralization, we assumed that the difference in $CO_2$ between microcosms with and without litter (for each treatment and water content) was entirely contributed by litter mineralization. Subsamples from soils collected at $T_{end}$ were extracted with 1 M KCl (1:5 soil:liquid ratio) for one hour, and the nitrate and ammonium concentrations were determined using a colorimetric method adapted to microplates[70,71].

### Microbial biomass

For microbial biomass C (MBC) analysis, subsamples from each of the six microcosms were randomly paired (e.g., jars 1&3, 2&4, and 5&6) and combined to form triplicate samples. MBC was measured following a standard fumigation-extraction method[72]. Briefly, one subsample (1.33 g dry wt. equivalent) was immediately extracted by agitating for one hour in $K_2SO_4$ (0.05 M, 10 mL), while the other subsample was fumigated for 24 hours with ethanol-free chloroform (0.4 mL) prior to extraction with $K_2SO_4$. The extracts were passed through filter paper (0.45 µm, Advantec) and dissolved organic C was measured by combustion catalytic oxidation (TOC-VCPN TOC analyzer Shimadzu, Japan). A portion of the extract was freeze-dried, and total C and $\delta^{13}C$ of the freeze-dried residue were measured using isotope ratio mass spectrometry (Delta V, Thermo Scientific, Germany) coupled to an elemental analyzer (NC2500, Carlo Erba, Italy).

### Soil organic matter fractionation

We modified a procedure[73] for fractionating soils according to size and density to characterize the effects of Ca on operationally defined soil organic matter fractions relevant to SOC cycling, namely: free and occluded particulate organic matter, and mineral-associated organic matter (fPOM, oPOM, and MAOM, respectively). fPOM and oPOM contain mostly plant matter at different degrees of decomposition, while MAOM contains silt- and clay-size primary particles and organo-mineral complexes as well as <53 µm microaggregates. Briefly,

subsamples were collected from each of the six microcosms prepared for each treatment. Sodium polytungstate (1.65 g cm$^{-3}$) was added to the soil samples to isolate the floating material (fPOM) after centrifugation and filtration. Next, soils were shaken end-to-end in fresh sodium polytungstate solutions (1.65 g cm$^{-3}$) for 18 hours with 3 mm glass beads to disrupt aggregates, and the floating material (oPOM) was collected. The remaining materials was shaken end-to-end in sodium hexametaphosphate (0.5% w/w) for 18 hours and sieved (53 μm) to separate sand from silt+clay sized particles (MAOM). All fractions were washed with deionized water, dried at 60 °C, weighed, and ball milled. The mean mass recovery was 102%; C recovery ranged from 96 to 109%, and N recovery ranged from 98 to 106%. Similarly to the microbial biomass samples, the six replicates were randomly paired and combined into triplicates samples. These were measured, along with bulk soils, for total C, N, $\delta^{13}$C, and $\delta^{15}$N, as above. We assumed total C equaled organic C since these soils did not contain carbonates, as previously determined[74].

## C K-edge and Ca L-edge near edge X-ray adsorption fine structure (NEXAFS) spectroscopy

The chemical composition of SOC and soil Ca was determined by NEXAFS which was acquired at the C K-edge (270 – 320 eV) and Ca L-edge (340 – 360 eV) on composited (n = 6) bulk soils and reference materials at the spherical grating monochromator (SGM) beamline of the Canadian Light Source (Saskatoon, SK). For C analysis, spectra were deconvoluted into Gaussian functions that constitute contributions from aromatic-C, substituted aromatic-C, alkyl-C, carboxyl-C, O-alkyl C, and carbonyl-C[75]. For Ca analysis, the position of the two main peaks (L$_2$ at 349.4 and L$_3$ 352.6 eV), and the L$_{2/3}$ split ratio, which reflects Ca crystallinity, were compared among soils samples and to a set of reference Ca-containing materials which were measured in parallel to the soil samples: calcite, Ca-smectite, CaCl$_2$, CaSO$_4$, Ca-acetate, Ca-formate, and Ca-citrate. Full details are provided in the Supplementary Information section.

## Midinfrared spectromicroscopy

We used FTIR spectromicroscopy to study the spatial distribution of organo-mineral associations. At least 8 sections (500 nm thick) of intact microaggregates (53–250 μm) were collected from samples composited for each treatment factor using a published protocol[76] and analyzed at the Mid Infrared Spectromicroscopy (Mid-IR) beamline of the Canadian Light Source (Saskatoon, SK) on an Agilent Cary 670 FTIR interferometer and an Agilent Cary 620 microscope. Full details on sample preparation, and data collection and analysis are provided in the Supplementary Information. Briefly, spectral maps (8-10 from each intact microaggregate) were created for the absorbance at 3620 (mostly O-H groups of aluminosilicates, as well as in aluminum hydroxides), 2920 (aliphatic-C), 1620 (aromatic-C), and 1413 cm$^{-1}$ (carboxylic-C). We acknowledge that the peak at 1620 cm$^{-1}$, defined here as aromatic-C[76–78], which includes the band spanning from 1477 to 1756 cm$^{-1}$, also contains peaks contributed by N-H bending, C = N stretching (-1545 cm$^{-1}$) and C = O stretching (-1650 cm$^{-1}$). We did not analyze the polysaccharide peak at 900-1290 cm$^{-1}$ since it overlaps with Si-O vibrations from clay minerals. The absorbance values were used to construct linear models to quantify the co-localization of organic chemical groups and clay minerals, which we assumed is driven by interactions. We interpret linear models as follows: 1) absorbance values reflect chemical abundance, 2) coefficients of determination (R$^2$) indicate co-localization, and 3) regression coefficients (β) quantify organic content per unit clay mineral. Due to beamtime limitations, only soils incubated at low water content at T$_{beg}$ and at T$_{end}$ were analyzed.

## NanoSIMS

A NanoSIMS 50 L (Cameca, Gennevilliers, France) was used to analyze the spatial distribution of litter-derived $^{13}$C$^{15}$N, $^{44}$Ca, and soil minerals

on intact aggregates. A total of 37 measurements (4-7 per treatment) were collected with the O$^-$ RF plasma source[79] to measure the spatial distribution of $^{23}$Na$^+$, $^{24}$Mg$^+$, $^{27}$Al$^+$, $^{39}$K$^+$, $^{40}$Ca$^+$, $^{44}$Ca$^+$, and $^{56}$Fe$^+$. The Cs$^+$ source was used to conduct 37 measurements (4-9 per treatment) to measure the spatial distribution of $^{16}$O$^-$, $^{12}$C$_2^-$, $^{13}$C$^{12}$C$^-$, $^{12}$C$^{14}$N$^-$, $^{12}$C$^{15}$N$^-$, $^{28}$Si$^-$, and $^{44}$Ca$^{16}$O$^-$. It is usual for NanoSIMS measurements to investigate the ion ratio $^{12}$C$^{14}$N$^-$/$^{12}$C$^-$ to (1) distinguish the resin from the organic matter when sections are analyzed, and (2) use it to characterize the composition of organic matter-dominated areas. N is measured as the cluster ion CN ($^{12}$C$^{14}$N$^-$) according to its improved ionization compared to $^{14}$N$^-$. For specific co-localization types identified by a machine learning-based classification, we calculated the $^{15}$N ratio ($^{12}$C$^{15}$N$^-$:($^{12}$C$^{14}$N$^-$ + $^{12}$C$^{15}$N$^-$)) to determine $^{15}$N enrichment, the normalized N:C ratio ($^{12}$C$^{14}$N$^-$):($^{12}$C$^-$ + $^{12}$C$^{14}$N$^-$), and the normalized Ca:N ratio ($^{40}$Ca$^+$ + $^{44}$Ca$^+$):($^{40}$Ca$^+$ + $^{44}$Ca$^+$ + $^{12}$C$^{14}$N$^-$ + $^{12}$C$^{15}$N$^-$).

We focused on $^{12}$C$^{15}$N$^-$ as a fingerprint for $^{15}$N-enriched litter since the high enrichment of litter-N (8.203 atom% $^{15}$N) compared to litter-C (1.895 atom% $^{13}$C) enabled more accurate isotopic analysis. Both primary Cs$^+$ and O$^-$ ion sources were tuned to match an imaging resolution of approximately 120 nm with a field of view of 30 μm at 256 pixels with 30-40 consecutive planes. The dwell time was 1 ms pixel$^{-1}$ at a primary current of 2 pA. Further details about sample preparation and image processing and segmentation are provided in the Supplementary Information.

## Bacterial community structure

Bacterial community structure was determined by 16 S rRNA gene amplicon sequencing. A full description is available in the Supplementary Information. In brief, DNA was extracted from 0.25 g of freeze-dried soil sample using the Qiagen PowerSoil kits, according to the manufacturer's protocol, and PCR was used to amplify the 16 S rRNA gene (V4 region; 515 f/806r). All sample metadata and raw sequencing data was archived under the NCBI BioProject accession: PRJEB48763 available at https://www.ncbi.nlm.nih.gov/bioproject/?term=PRJEB48763. Sequencing data was processed using QIIME2 (v. 2020.2)[80] with a dependency on DADA2 (v. 1.10)[81] to assign sequences to amplicon sequence variants (ASVs). Taxonomic classification was performed using the QIIME2 'q2-feature-classifier' trained on the Silva database (v. 138)[82]. All counts were normalized by the proportion of total reads and presented as counts per thousand reads.

## Data analysis and statistics

The proportion of litter-derived C and N in soil and soil fractions, and C in microbial biomass C and CO$_2$, was determined from isotope data by applying a two-source mixing model,

$$f_{litter} = \left( \frac{at\%_{sample} - at\%_{control}}{at\%_{litter} - at\%_{control}} \right) \quad (1)$$

where, $f_{litter}$ is the proportion of litter-derived C or N in a sample, $at\%_{sample}$ is the atom% $^{13}$C or $^{15}$N value of a sample (incubated with litter), $at\%_{control}$ is the atom% $^{13}$C or $^{15}$N value in the control sample (incubated without litter) and $at\%_{litter}$ is the atom% $^{13}$C or $^{15}$N of the labelled litter. $f_{litter}$ was multiplied by C or N concentration in each pool to calculate litter-derived C or N in μg C g$^{-1}$ soil or μg N g$^{-1}$ soil.

$f_{litter}$ in microbial biomass was calculated as:

$$atom\%^{13}C_{mic} = \frac{at\%^{13}C_f * C_f - at\%^{13}C_{nf} * C_{nf}}{C_f - C_{nf}} \quad (2)$$

Where $at\%^{13}C_f$ and $at\%^{13}C_{nf}$ are the $at\%^{13}$ values of the fumigated and non-fumigated samples and $C_f$ and $C_{nf}$ are the extracted C contents (μg C g$^{-1}$ soil) of the corresponding samples. The control and treatment values for $atom\%^{13}C_{mic}$ were incorporated into the two-source mixing model to calculate the fraction of litter incorporated into the microbial

biomass, then multiplied by the MBC pool size ($\mu g$ MBC $g^{-1}$) to calculate the mass of substrate incorporated into the biomass (M$B^{13}$C).

Microbial carbon use efficiency (CUE) was calculated as:

$$CUE = \frac{MB^{13}C}{MB^{13}C + {}^{13}CO_2} \qquad (3)$$

All statistical analyses were performed in R v4.2.0. The effects of Ca treatment, water content, and sampling time, and their interactions on cumulative $CO_2$ mineralization, microbial biomass C and CUE, total C and N, and proportion of labeled litter in $CO_2$ and soil fractions, were evaluated with an ANOVA using the *aov* function. Differences among treatments were evaluated using the *emmeans* function. Cohen's D effect size estimates were calculated using differences of estimates between treatments using the *eff_size* function available in the *emmeans* package. Linear regressions between peak heights obtained from FTIR spectromicroscopy were done using the *lm* function. We then applied ANCOVA tests to compare linear regression coefficients for each predictor-response pair (i.e., clay mineral-functional group) between Ca-treated and control soil for each time point, and between $T_{beg}$ and $T_{end}$ for Ca-treated and control soils, by constructing linear models which include an interaction term (clay*treatment or clay*time, respectively). A significant interaction term in the ANCOVA model indicates that the effect of the predictor (clay mineral abundance) on the response variable (organic function group abundance) differs significantly between groups.

The alpha- and beta-diversity of bacterial communities were analyzed using the following approaches. PERMANOVA testing was performed using 'adonis' from the *vegan* package (v. 2.5.7) based on the Bray-Curtis dissimilarity calculated from bacterial community composition. The differential abundance of ASVs between contrasting treatments (±litter, ± calcium, and high / low moisture) were identified using the R package *DESeq2* (v. 1.30.1). Non-significant indicator ASVs ($p_{adj} > 0.05$) and those which were less than 3-fold differentially abundant were excluded from analyses. The similarity among bacterial communities was visualized using t-distributed stochastic neighbor embedding (t-SNE) with *Rtsne* (v. 0.15). Significant effects are denoted by asterisk: $p < 0.05$ (*), $p < 0.01$ (**), and $p < 0.001$ (***). Analyses were performed in R with a general dependency on the following packages: *phyloseq* (v. 1.34)

### Reporting summary

Further information on research design is available in the Nature Portfolio Reporting Summary linked to this article.

## Data availability

All data required to reproduce the manuscript results are available at Zenodo: 10.5281/zenodo.8139855. A complete list of amplicon sequence variants (ASVs) is provided in Supplementary Dataset 1. All raw sequencing data and associated metadata was archived under the NCBI BioProject accession: PRJEB48763 available at https://www.ncbi.nlm.nih.gov/bioproject/?term=PRJEB48763.

## Code availability

The codes required to reproduce the results are available at Zenodo: https://doi.org/10.5281/zenodo.8139855.

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

## Acknowledgements

This work was supported by BARD, the United States - Israel Binational Agricultural Research and Development Fund, Vaadia-BARD Postdoctoral Fellowship Award No. FI-573-2018 (I.A.S), US National Science Foundation (IOS-2034351) (J.L), and the U.S. Department of Energy, Office of Biological & Environmental Research Genomic Science Program Award No. DE-SC0016364 (D.H.B). Part of the research described in this paper was performed at the Canadian Light Source, a national research facility of the University of Saskatchewan, which is supported by the Canada Foundation for Innovation (CFI), the Natural Sciences and Engineering Research Council (NSERC), the National Research Council (NRC), the Canadian Institutes of Health Research (CIHR), the Government of Saskatchewan, and the University of Saskatchewan. We thank the Genomics Facility (RRID:SCR_021727) of the Biotechnology Resource Center of Cornell Institute of Biotechnology for their help with sequencing experiments.

## Author contributions

I.A.S. drafted and wrote the manuscript, designed and conducted experiments, and collected and analyzed data; R.C.W. wrote the manuscript, collected and analyzed data, and provided critical revision of the article; and S.A.S. collected and analyzed data, and provided critical revision of the article; C.H. collected and analyzed data; D.H.B. supported data collection and provided critical revision to the article; J.L. aided in designing the experiments, and provided critical revisions to the article. All authors have approved the final version for publication.

## Competing interests

The authors declare no competing interests.
