## [Peer Review File · Nature Communications]

Calcium promotes persistent soil organic matter by altering microbial transformation of plant litterReviewer #1 (Remarks to the Author):

This interesting study proposes that the increase in SOC commonly observed in Ca-rich soils may be driven in part by Ca-mediated shifts in microbial communities and carbon-use efficiency, which lead to increased formation of mineral-associated organic matter. This proposal contrasts with the conventional explanation for this phenomenon, which is based on physico-chemical protection alone. In my opinion, the authors provide convincing evidence for some, but not all, aspects of this framework.

Several findings do not seem to fit with the conceptual model articulated by the authors, including the treatment responses of MAOM C at the beginning and end of the experiment. At the beginning, MAOM-C from litter is actually lower in the Ca treatment than the control (no explanation is provided). At the end, the values appear to be equivalent in Fig 1 (although Fig S5 indicates a likely interaction with moisture treatment). According to the central argument of the manuscript, I would expect MAOM-C to be higher in the Ca treatment in both cases. Also importantly, the NanoSIMS data seems to focus exclusively on ^{15}N , not ^{13}C , as a metric of MAOM formation from litter (the litter included both isotope labels). The reader needs to also see the C data to evaluate whether Ca is actually driving MAOM formation via the proposed mechanism, especially since not all of the ^{15}N observed in the images is necessarily in organic form. After a 4-month incubation, there should be significant ammonium or nitrate N from N mineralization/nitrification associated with ion exchange sites, which is difficult to distinguish from organic N in the NanoSIMS images). Could greater N mineralization explain the difference in C:N of MAOM in the Ca addition treatment?

Linear regressions based on the pairwise spatial relationships between clay minerals and various organic moieties determined by spectromicroscopy are interpreted as evidence for enhanced formation of organo-mineral interactions in the Ca treatment (Fig 4), but I don't believe that appropriate statistical methods were used here. Strictly speaking, we would want to see a statistical test for differences in regression slopes between the Ca and control treatment for each pair of predictor/response with data from the end of the experiment (i.e., an ANCOVA approach), but we are not given those tests; rather, we are just presented with numerical estimates of slopes, with no metric of variability in the slope estimate, nor tests of significant differences among slopes. There may well be statistical evidence for differences in some of these cases. However, the largest visible difference seems to occur for aromatic C, which is at odds with the claim that the newly formed MAOM is entirely of microbial necromass origin. In general, I would ask the authors to temper the claim that "MAOM is comprised mainly of microbial products" (L44), as we know that plant-derived DOC can be an important component in many cases (e.g. Angst et al. 2021, 10.1016/j.soilbio.2021.108189), particularly in situations where fresh litter, rich in soluble C, is added to a soil with significant capacity to sorb DOM. In fact, a more nuanced view of MAOM sources may help resolve some of the discrepancies in the dataset noted above.

I was also surprised that the authors entirely ignored the potential role of fungi in driving the observed responses to Ca addition, focusing exclusively on bacteria. Ca is an important constituent of fungal cell walls and has been shown to increase growth of white rot fungi, which can degrade lignin. This needs to at least be acknowledged and incorporated in the framework, even if fungal sequence data are not available. Fungi are important because if Ca stimulates fungal growth and litter decomposition (as observed in several other studies, some cited in the Introduction; discussed in Lovett et al. 2016), we would need to alter the conceptual framework in Fig 6. In situations where white rot fungi are promoted by Ca, this would actually select for the "fast, inefficient" decomposers that are purportedly suppressed by Ca addition (Fig. 6). There are other aspects of the conceptual framework that don't appear supported by the data, e.g. feedbacks among microbial colonization and MAOM formation (described in the detailed comments below).

Throughout the manuscript, the reader is presented with p-values, but seldom (if ever) with effect sizes. In many cases, the apparent effect sizes are very small (e.g., the difference in cumulative C mineralization between Ca and control treatments in Fig 1 A). Given the small effect size for this and several other metrics, and the absence of an observable change in new MAOM from litter as a function of Ca addition (according to ^{13}C), I am a little skeptical about the overall importance of the proposed mechanism. It would help to compare the absolute magnitudes of the C pools/fluxes under the different treatments to more rigorously establish this context, or at least to establish

that the variability among replicates precludes precise detection of certain effects predicted by the conceptual model.

Specific comments:

L44: suggest different word than "mainly", in light of conflicting evidence in the literature

Rationale for "water content" treatments needs to be introduced before the Results

L106: Fig S6 shows that DOC was higher in the Ca treatment at high water content at the beginning of the experiment. These interactions are important and challenge the simple interpretation of main effects.

L138: "The effects of Ca on microbial community structure and utilization of litter affected bulk SOC composition" your data don't conclusively support the causal nature of this statement

L147: what kinds of clay minerals, or are you referring to total clay-sized particles? This is difficult to interpret. I am not certain what this metric is telling us.

L161: What about ^{13}C ? Since the litter was ^{13}C labelled, this would seem to be another crucial element of the argument

L168: I don't understand precisely what you mean by "co-localization of Ca with organic matter regions", how can this be quantified as a percentage? A little more description needed; would seem to be heavily determined by the methods used for selecting regions of interest from the NanoSIMS data.

L172: Is "CN:C" a typo (do you mean N:C ratio? Or if not, more concrete explanation and rationale is needed for this metric)

L178: what is "Ca:CN", is CN this sum of C and N, or something else? Also very confusing in the figures

L219: This statement is apparently based on the ^{15}N NanoSIMS data, but in principle some of observed N associated with minerals could have been NH_4^+ mineralized from litter, or NO_3^- (could constrain this potential contribution using reasonable estimates of N mineralization based on stoichiometry). More importantly, if the argument holds, it could be bolstered by reporting the ^{13}C NanoSIMS data.

L274: please provide further details of the agroecosystem management. What crop type, how was it managed, fertilization history, etc.

L278: the $\delta^{13}\text{C}$ is wrong (negative sign needed)

L319: I don't see why you need to make this assumption—since you are measuring CO_2 and $\delta^{13}\text{C}$, you can calculate litter decomposition directly. Or is this merely a heuristic for deciding when to end the incubation, based on the KOH data?

L347: I don't know what you mean by "six replicates were randomly paired"—what is being paired here, presumably each replicate is measured separately (if not, why not)

L435: I think you mean $p_{\text{adj}} > 0.05$ for non-significant ASVs?

L437: I don't understand your linear models—are you modeling each ASV, or some multivariate index? Please provide more details

Fig 1B : The $P < 0.001$ appears to refer to the test comparing the low/high water content, but this is not of interest for the main question (Ca effect on litter decomposition); please clarify if there were differences between the Ca/control at each water content

Fig. 1E,F: In my mind, these panels are misleading because they obscure the importance of the time x water interaction that you show elsewhere.

Fig 3: In my opinion, the Ca vs control spectra should be plotted on top of each other, with different time points shown in panels. We care more about the Ca effect than the time effect, and it is impossible to discern the Ca effect with the current layout

Fig 4: In my view the statistics are not appropriate for the question at hand. What we care about is whether the slopes of these regressions significantly differ between the Ca and control treatments, not their absolute values. I suspect that there may not be statistical evidence for different slopes in several of these panels.

Also, the text in panels b,c,d seems incorrect (check the colors)

Fig 5: I think there is a mistake, there is no Ca:CN shown

692: I don't understand the rationale for plotting (^{12}C ^{14}N):(^{12}C - + ^{12}C ^{14}N -), this needs to be clearly articulated

Fig. 6: what is the experimental evidence for part e. of the diagram? How can it be argued that the cycle intensifies over time? If this is actually correct, then I would assume that the new surface-colonizing bacteria would also be consuming some of the "MAOM", which might be consistent with the absence of observed increase in MAOM ^{13}C in this experiment

Fig. 7 should be placed first, as Fig 1, to introduce the reader to the experiment

Table S2: clarify the contrast control-Ca; does a negative t ratio imply a greater value in the control, or Ca treatment?

Figure S5: It is misleading to draw boxplots for only three reps per treatment, please remove the boxplots and just show the points.

Reviewer #2 (Remarks to the Author):

Calcium promotes persistent soil organic matter by altering microbial transformation of plant litter.

In this manuscript the authors completed microcosm experiments on a Mollic Endoaqualf silt loam soil collected from a fallow subplot in Ithaca. The authors added $^{44}\text{CaCl}_2$ / CaCl_2 and $^{13}\text{C}^{15}\text{N}$ labelled willow leaves to different microcosms, destructively sampling after 4 days and ~4 months. The authors measured changes in microbial biomass, litter incorporation, community composition, carbon use efficiency, and respiration rates. They also completed standard characterization, spectroscopic measurements (bulk C/Ca NEXAFS; FTIR spectromicroscopy), density fractionation, and NanoSIMS measurements on their microcosms. The authors demonstrate that Ca addition reduced respiration rates by a small amount relative to the control (5%), influenced microbial community composition, carbon use efficiency (40%), and increased incorporation of litter into microbial biomass by 20%. Through the NanoSIMS and Ca labelling work the authors also novelly demonstrate ^{44}Ca enrichment and increasing co-localization of Ca with organic matter (68-90%) upon Ca addition. This work is novel and furthers our understanding of the role of Ca in SOC dynamics, incorporating its influence on microbial communities.

I think that this manuscript could be published in Nature Communications once the authors address several key major revisions. In their discussion, the authors stick closely to a single narrative, but I believe they could instead explore a range of potential explanations of their data and be more cautious in their interpretations. Particularly when their dataset is in disagreement with other Ca incubation studies^{18,25}. A lot of the interpretations seem to hinge on the increased incorporation of litter derived C into the mineral associated organic matter pool, yet unless I'm mistaken, the percent of ^{13}C incorporated to the MAOM pool from litter was lower in the Ca

addition sample (T beg.) before being equal (if not slightly less) (Fig. 1E). Furthermore, I don't feel as though the effects of the pre-incubation / leaching step or KCl control data are adequately covered in the discussion, and as such, will need to be incorporated more clearly. I would urge the authors to be cautionary in their interpretations and language, exploring all potential influences of their methods and causes of their results in the discussion. I thank the authors for moving the field forward and hope that they will submit a revised manuscript to Nature Communications.

Major comments

- Please report and discuss all T beg. and T end. Results. While Ca addition affected CUE and ^{13}C transferred from litter to microbial biomass at T Beg., it had no effect at T End? This data should be displayed in the main graph and this result should be more clearly discussed. The results at T beg. are discussed in some locations and T End in others. If we are to infer any conclusions about the short to medium-term effects of Ca addition on SOC dynamics, both observations should be reported and explained clearly (Fig. 1 in particular).
- Please report the effects of Ca addition on the MAOM fraction clearly. Line 107 (also line 180-183, 219-221) states that Ca influenced the transfer of litter into the MAOM fraction, which to the reader, implies that Ca probably increased the incorporation of ^{13}C from litter into the MAOM fraction. Yet, in your results it actually seems that at T Beg. (Fig. 1E) there was less C being incorporated to MAOM in the Ca addition samples, seemingly Ca addition reduced MAOM incorporation until T End when there was no difference. This needs to be made far clearer in this section and throughout the manuscript, see some other locations above.
- Please report all of your density fractionation data. Following on from my last comment, according to line 158 your spectromicroscopy and line 168 NanoSIMS results indicate increased formation of organo-mineral association because of Ca addition, but this was not from the litter as per Fig. 1E. How did your density fractionation results change? Why aren't these reported? Maybe I missed them, but I couldn't find them in your results or SI. You mention on Line 346 that six replicates were randomly paired and measured, but I don't see the results from the sample mass, C, N, $\delta^{13}\text{C}$ or $\delta^{15}\text{N}$ of each fraction? This seems like it would be an easy way to quantify an increase in mineral associated SOC caused by Ca addition? Your recoveries also seemed high for your density fractionation (109%), was the SPT adequately washed out from these fractions? Did you measure the EC of your rinse solution?
- How sure are you that the addition of Ca selected for these microbial taxa because of their survival strategy as surface adhering organisms, and thus, their ability to excrete substances, such as extra-cellular polysaccharides, for which Ca is a critical co-factor? How sure are you that these taxa were only selected for because of their supposed ability to adhere to surfaces and not, for instance, their ability to tolerate stress, be versatile, or have a range of functional traits that benefit from Ca addition? Furthermore, I believe that actinobacteria which form mycelia also coat themselves in Ca oxalate as a strategy. Could you rule out other explanations for your data? If it was purely due to the increased excretion of surface-adhering compounds, then why do you see a decrease in your O-alkyl C content in the Ca addition treatment? Wouldn't this region of C K-edge spectra (289.4 eV) increase if there was more abundant excretion of these compounds by organisms adhering to surfaces? If Ca was instead stressing the microorganisms or causing them to succeed due to other strategies, this could also explain your observations? This needs to be addressed properly in the discussion. I would urge the authors to be more cautious in their interpretations and present all possible explanations of these changes in microbial community composition.
- It seems that the KCl experiments are extremely important as a control? Why were they not run for the same extended period of time? Furthermore, what was the effect of KCl addition on the microbial community composition? Is it possible that the changes that you have recorded could just be an effect of increasing ionic strength on the microbial community? Please report all available data on the KCl control and discuss it clearly in the discussion.
- Line 232 Do you believe that your DOC measurements could have also been affected by your pre-treatment? It seems that you measured the DOC in your MBC analysis and not in the leachate of your pre-incubation step. Yet, you also pre-incubated and then leached your soils? What would be the effect of this on DOC content? What if the DOC (that had not been bound to Ca in your

control) had already been flushed from your samples during this step? Was this measured? Are you sure that DOC export was not reduced like has been seen in other studies¹⁸, particularly at circumneutral pH²⁵, like your own study? Why was this pre-incubation step taken? and please discuss it more clearly in your discussion.

- Line 247 This is an interesting conclusion because in Supplementary Table 1, the main changes to your exchange complex are a reduction in Mg, another divalent cation. The monovalent cations Na and K barely change between your treatments. Thus, if Ca is replacing Mg on the exchange complex, why is this creating new sites for microbial attachment? Wouldn't you expect these sites to be somewhat similar due to the divalency of both cations? Please explore why this is the case in your discussion and suggest all possible explanations of your data, being more cautious in your interpretations.

Minor comments

Line 22 Replace increase with contributes to.

Line 29-31 The meaning of this sentence is convoluted by its length. I'd suggest breaking up the arguments.

Line 36 Exchangeable Ca can also be found on the exchange sites of organic matter and other negatively charged soil minerals, not solely on clay minerals.

Line 46 and physically protected within the soil matrix?

Line 49 soil mineralogy also?

Line 74-75 Shouldn't this be a separate point?

Line 83 I don't believe that bulk Near Edge XAFS requires capitalization in its full form here.

Line 166 There is an extra unbalanced bracket here.

Line 166-167 This sentence probably needs to come before Line 164 as it sets up the preceding argument.

Line 168-170 How precise are these percentages? Depending on their precision, I'd suggest reporting these percentages to the nearest whole number?

Line 195 – 198 Overall Ca addition selected against the communities that were positively enhanced by litter addition and reduced the incorporation of this litter to MAOM at T. Beg.?

Line 225 Rowley et al., 2021 <https://doi.org/10.1007/s10533-021-00779-7> specifically supports your density fractionation observations regarding the importance of the mineral-associated pool and reduced importance of the occluded fraction as per line 111 to 118.

Line 226 Does this not also show that Ca is contributing to the preservation of plant-like organic compounds that are rich in aromatic and carboxylic moieties?

Line 232 This is contrary to the results of Minick et al., 2017 <https://doi.org/10.1007/s10533-017-0307-z> your reference 18 and should be discussed more clearly. As you mention in your intro, Minick saw a reduction in the mineralization of SOC and enhanced mineralization of litter. This could also be covered more clearly in your results. What do you think the effect of your pre-incubation could have been? Could this have also influenced your DOC results?

Line 253 Doesn't the higher proportion of aromatic C moieties indicate that this C is slightly less processed than your control soil?

At current it is not clear how many replicates were made in the different treatments? Was it 6 for each treatment? I believe so, but it is not very clear as currently worded in the methods. Including the number of observations throughout, even the figure captions, would be useful for the reader and more clearly delineating this in the methods section will also be important.

What was the influence of your pre-incubation step? Why was this taken? Did you measure changes in microbial community at this stage also before leaching your soils, as part of a disturbance control? Did you also measure the effects of KCl on microbial community composition? Please report the KCl data more clearly.

Figures

Please present T End also in Fig. 1. or consider breaking this up into multiple figures. Consider moving the explanation of the error bars for A up to the correct location (I was about to enquire about what the error bars meant).

Supplementary Fig 2, 3. While the Ca L-edge has not shifted, I'm not sure that this lends credence to the idea that Ca addition did not affect Ca speciation. I think this would be hard to see at the Ca L-edge, which is less detailed than at the Ca K-edge. I'd suggest being more careful with your interpretations here.

Supplementary Fig 2, 3. Is there also a normalization issue with some of your standards (Ca L-edge) particularly the calcite, citrate, and formate?

Tables

Please add all available data, including the density fractionation work.

Please add a table detailing the effects of the pre-incubation step on your soils, prior to the start of your experiments.

Response to Reviewer Comments for Shabtai et al., "Calcium promotes persistent soil organic matter by altering microbial transformation of plant litter"

Please note that we have updated all taxonomic classifications in the article to the most current nomenclature implemented by the National Center for Biotechnology Information (ex. *Actinobacteria* are now *Actinomycetota*).

In response to Reviewer #1's suggestion, we reordered Figure 7 as Figure 1 to introduce the reader to the experimental design at the beginning of the paper and renumbered all figures accordingly. The new figure numbers are used in the reply to comments below.

Response to Reviewer #1:

This interesting study proposes that the increase in SOC commonly observed in Ca-rich soils may be driven in part by Ca-mediated shifts in microbial communities and carbon-use efficiency, which lead to increased formation of mineral-associated organic matter. This proposal contrasts with the conventional explanation for this phenomenon, which is based on physico-chemical protection alone. In my opinion, the authors provide convincing evidence for some, but not all, aspects of this framework.

We appreciate your interest and constructive critique of our manuscript. We hope that our revisions have addressed your concerns about the evidence supporting our claims. Please find a point-by-point response to your comments and suggestions below. We appreciate your feedback, which has improved the quality of our manuscript.

Main comments

1. Several findings do not seem to fit with the conceptual model articulated by the authors, including the treatment responses of MAOM C at the beginning and end of the experiment. At the beginning, MAOM-C from litter is actually lower in the Ca treatment than the control (no explanation is provided). At the end, the values appear to be equivalent in Fig 1 (although Fig S5 indicates a likely interaction with moisture treatment). According to the central argument of the manuscript, I would expect MAOM-C to be higher in the Ca treatment in both cases.

Thank you for highlighting this issue. We have clarified these points in the Discussion (L261-270).

"In the Ca-treated soils, a larger fraction of MBC at T_{beg} was derived from litter-C than in control soils, indicating that more litter was cycled through microbial biomass, while concomitantly less MAOM from litter was measured in Ca-treated soils. This is likely an observation of a transient nature at T_{beg} which captures the greater uptake of litter-C into MBC prior to its conversion into MAOM. At T_{end} , we found that the amount of litter-derived C and N transferred to MAOM was not different (Fig. 2E), which may be explained by the fact that surface-adhering bacteria likely also consumed litter-derived MAOM throughout the incubation. The observation that the proportion of labeled C in MBC was significantly higher after Ca addition even at T_{end} (Fig. 2D and Supplementary Table 2) supports this claim. So does the observation at T_{end} that MAOM consisted of compounds that were more microbially processed (lower C:N) after Ca addition (Fig. 2F). Thus, MAOM composition, but not amount, was impacted by Ca addition."

Confirming the role of microbial processing in conversion of plant C into MAOM C is the observation that at T_{end} , after 4 months without any additional exogenous C input, MAOM-13C was higher at high water content. We believe this is due to the greater microbial activity at higher water content, which increased conversion of litter-C to MAOM-C (Figure 2E).

We also propose explanations for why isotope/elemental analysis did not find differences in MAOM between Ca and control, while FTIR microscopy did show differences (L283-288). We wrote:

"The discrepancy between FTIR-microscopy analysis which indicated greater abundance of organic functional groups in Ca-treated soils and elemental and isotope analysis that did not show differences (Fig. 1E, Supplementary Fig. 7) could stem from the spatially resolved nature of the former vs bulk characterization of the latter, and from the fact that FTIR-microscopy was done on microaggregate (53-250 μm) sections while the isotope analysis was done on > 53 μm particles."

2. Also importantly, the NanoSIMS data seems to focus exclusively on 15N, not 13C, as a metric of MAOM formation from litter (the litter included both isotope labels). The reader needs to also see the C data to evaluate whether Ca is actually driving MAOM formation via the proposed mechanism, especially since not all of the 15N

observed in the images is necessarily in organic form. After a 4-month incubation, there should be significant ammonium or nitrate N from N mineralization/nitrification associated with ion exchange sites, which is difficult to distinguish from organic N in the NanoSIMS images). Could greater N mineralization explain the difference in C:N of MAOM in the Ca addition treatment?

We agree that ^{13}C analysis by NanoSIMS would have provided confirmatory and complementary information; however, as stated in the Methods section (L484-486), we could not identify hotspots with ^{13}C enrichment that would allow using it to trace the transformation of litter-derived OM through NanoSIMS measurement. This was due to the lower enrichment of ^{13}C than ^{15}N in our litter.

Thank you for these helpful thoughts regarding N species detected with NanoSIMS! Following your comment, we extracted the soils collected from the incubation experiment at T_{end} and quantified nitrate and ammonium N in 1M KCl extracts (added to Methods L404-406). We found that inorganic N constituted up to ~1% of total N (provided in Supplementary Fig. 16 and below), which matches expectation of usually a low proportion of inorganic N as a fraction of total N. In addition, ^{15}N and ^{14}N were detected based on the CN^- signals which are mostly from organic N compounds (Li et al., 2016 <https://doi.org/10.1021/acs.analchem.5b04740>, ref #33 in manuscript).

Based on these two points we now added to the results section (L203-205)

"The contribution of inorganic-N species to this result is negligible since they made up, at most, ~1% of total N (Supplementary Fig. 16), and in addition, the fragment used for N detection (CN^-) is mostly contributed to by organic N compounds³³."

We did find differences in nitrate and ammonium concentrations between treatments which are generally consistent with our other results, as we explain below. While these interesting findings support the idea that Ca impacted OM cycling in our soils, these specific results are outside the scope of this current manuscript.

We write in the SI following Fig S16:

"These results show that total inorganic N species constituted up to ~1% of total N (ca. 0.34 mg N/g soil) and therefore were of negligible importance to our NanoSIMS results. Notably, differences between treatments were observed especially for nitrate N but also for ammonium N. Higher nitrification in Ca-treated soil and higher water content is consistent with greater microbial activity. Lower ammonium concentration in Ca-treated and high water content soils can be explained by a slower rate of ammonification compared to nitrification. In that case, although ammonification increased, ammonium still decreased."

Regardless of the proportion of inorganic vs organic N detected with NanoSIMS, all ^{15}N enriched compounds are litter-derived. Thus, it does not impact our conclusions if, at a given time point, some litter-derived N compounds were mineralized or nitrified. It is reasonable to believe that shortly before or after that time point some of these compounds will be immobilized, and so forth.

Supplementary Fig. 16. Concentration of nitrate N (A) and ammonium N (B) in calcium-treated and control soils incubated with labeled litter at low or high water content and sampled at T_{end} . Five to six replicated microcosms from each treatment combination were measured.

3. Linear regressions based on the pairwise spatial relationships between clay minerals and various organic moieties determined by spectromicroscopy are interpreted as evidence for enhanced formation of organo-mineral interactions in the Ca treatment (Fig 4), but I don't believe that appropriate statistical methods were used here. Strictly speaking, we would want to see a statistical test for differences in regression slopes between the Ca and control treatment for each pair of predictor/response with data from the end of the experiment (i.e., an ANCOVA approach), but we are not given those tests; rather, we are just presented with numerical estimates of slopes, with no metric of variability in the slope estimate, nor tests of significant differences among slopes. There may well be statistical evidence for differences in some of these cases.

Thank you for this constructive suggestion which prompted us to perform ANCOVA tests on each organic group to determine the significance of the difference in predictor-response slopes between Ca and control at the beginning and end of the experiment, and between T_{beg} and T_{end} for each treatment. We added these values to Supplementary Tables 7 and 8. Description of the statistical approach was added to the Methods section (L529-535).

Briefly, we found that for each of the functional groups, the slopes found for Ca were always significantly higher than for control, at both T_{beg} and T_{end} . The greatest difference in slope between T_{beg} and T_{end} for Ca was in carboxylic and aromatic groups, while a negative difference was found for aliphatic. The greatest difference in control soil was for aliphatic and aromatic, while the smallest was for carboxylic. We believe these findings provide additional evidence to support the claim that Ca addition resulted in formation organo-mineral associations with more microbial-like compounds.

However, the largest visible difference seems to occur for aromatic C, which is at odds with the claim that the newly formed MAOM is entirely of microbial necromass origin. In general, I would ask the authors to temper the claim that "MAOM is comprised mainly of microbial products" (L44), as we know that plant-derived DOC can be an important component in many cases (e.g. Angst et al. 2021, 10.1016/j.soilbio.2021.108189), particularly in situations where fresh litter, rich in soluble C, is added to a soil with significant capacity to sorb DOM. In fact, a more nuanced view of MAOM sources may help resolve some of the discrepancies in the dataset noted above.

The largest difference with time for Ca was for carboxylic (Supplementary Table 8).

We rephrased L49, as well as L185-186 and L293-295 to acknowledge the fact that MAOM can also include plant derived compounds and referenced Angst *et al.* 2021.

4. I was also surprised that the authors entirely ignored the potential role of fungi in driving the observed responses to Ca addition, focusing exclusively on bacteria. Ca is an important constituent of fungal cell walls and has been shown to increase growth of white rot fungi, which can degrade lignin. This needs to at least be acknowledged and incorporated in the framework, even if fungal sequence data are not available. Fungi are important because if Ca stimulates fungal growth and litter decomposition (as observed in several other studies, some cited in the Introduction; discussed in Lovett et al. 2016), we would need to alter the conceptual framework in Fig 6. In situations where white rot fungi are promoted by Ca, this would actually select for the "fast, inefficient" decomposers that are purportedly suppressed by Ca addition (Fig. 6). There are other aspects of the conceptual framework that don't appear supported by the data, e.g. feedbacks among microbial colonization and MAOM formation (described in the detailed comments below).

It should be noted that microbial biomass measurements include fungi and bacteria and therefore carbon use efficiency calculations capture the activity of both. The relative extraction efficiency, however, is not clear. In addition, any of the spectroscopic results that reflect differences in C composition and spatial distribution are also likely dependent on fungal as well as bacterial activity. We therefore do not feel that we entirely ignored the role of fungi in our work.

However, we completely agree that fungal community composition data would have been a nice addition to our experiment, without which we cannot evaluate their importance in the observed phenomenon. We have now acknowledged their potential importance in the Discussion (L240-244) and have also highlighted the potential role of calcium on fungal populations in the Introduction, writing "The microbial processing of plant C can also be impacted by soil Ca content, since Ca is a key factor in the growth and activity of fungi and bacteria, in particular surface-adhering

and biofilm-forming bacteria, as well as fungal lignin-degrading enzymes (24)" (L54-58). We did not focus our sequencing efforts on fungi for a few reasons: bacteria tend to dominate litter decomposition in the mineral soil (the focus of our study), while fungi dominate the decomposition of surface litter (see ref 47), and bacteria tend to dominate in agricultural soils due to the alkaline pH, and higher frequency of disturbance (at least compared to the hyphal mats that form in forest soils). White rot fungi, in particular, are not typically abundant in agricultural soils, and are generally outpaced by the growth of yeast-like fungi. Regardless, the major shifts in bacterial populations we observed confirmed our expectation of their importance in our study system and represent a novel view of the impacts of Ca on the soil microbiome.

5. Throughout the manuscript, the reader is presented with *p*-values, but seldom (if ever) with effect sizes. In many cases, the apparent effect sizes are very small (e.g., the difference in cumulative C mineralization between Ca and control treatments in Fig 1 A). Given the small effect size for this and several other metrics, and the absence of an observable change in new MAOM from litter as a function of Ca addition (according to 13C), I am a little skeptical about the overall importance of the proposed mechanism. It would help to compare the absolute magnitudes of the C pools/fluxes under the different treatments to more rigorously establish this context, or at least to establish that the variability among replicates precludes precise detection of certain effects predicted by the conceptual model.

Thank you for these constructive comments. We have made the following changes following your recommendations:

1. To Supplementary Tables 2-5, we added the estimate of the difference in means for Control-Ca. The estimate takes on the units of the tested variable, i.e., unitless for carbon use efficiency, and mg CO₂/g SOC for mineralizability). This provides absolute magnitudes of C pools/fluxes. We also now calculate and report the effect size (using Cohen's d) for each variable. Briefly, Cohen's d calculates the effect size by comparing the difference of the means divided by the pooled standard deviation. Effect sizes are unitless. A widely accepted approach is to consider values of 0.2, 0.5, and 0.8 as small, medium, and large effect sizes.
2. In addition to P values we added information on the magnitude of change (usually in % difference) between treatments. (L104, L116-118, L125, L129).
3. The 4% difference in CO₂ observed in our short-time incubation (~4 months) is indicative of an overall effect of Ca addition on respiration in this controlled experiment. A field experiment would likely yield more variable results. Statistically, this difference is significant (P = 0.006 averaged across both water contents), and the effect size is considered large (Cohen's d = 1.05). We attempted to compare this effect size to several other studies which have tested the effects of calcium additions on soil and litter mineralization in lab incubations and reported effect size.
 - Minick et al., (2017, Biogeochemistry), incubated soils from the Hubbard Brook plots which received 850 or 4250 kg Ca/Ha as wollastonite either with or without labeled maple leaf litter for ~1 year. They report a 1.7% and 7% increase in leaf litter respiration in the soils receiving low and high application rates, respectively, which is comparable to our results considering our incubation **lasted 1/3 as long**. Without litter addition, they report a 6% increase and 22% decrease in CO₂ for low and high rates, respectively. In our litter-free mesocosms which were used for the mixing model, we also found a ~17% reduction in CO₂ following Ca treatment (not shown in the paper). It should be noted that Minick et al. studied a forest soil with an initial pH of 4.1 and that wollastonite increased pH, which was associated with changes in respiration.
 - Similarly, Inagaki et al. (2017, Science of the Total Environment, ref #32), applied lime and incubated a highly weathered soil and reported a 4.1% difference in respiration compared to the control.
 - Other relevant studies either found no difference (Marinos and Bernhardt 2018, Ecology, ref #30), did not conduct a comparable lab incubation (Lovett et 2016, Ecosystems, ref #19), or did not report effect size or absolute difference (Whittinghill and Hobbie 2012, Biogeochemistry, ref #26).
4. Also, we found that in the process of saving the original Figure 2A from R to Adobe Illustrator, the graphics were distorted because of the way the aspect ratio was defined. The result of this was that the data points seemed too close which gives a visual impression of insignificant difference. The new plot shows that difference better. It is the same data and the same statistical significance, but visually it is more accurate.

Specific comments:

L44: suggest different word than “mainly”, in light of conflicting evidence in the literature

We rephrased this sentence to reflect a more nuanced view of plant vs microbe contribution to MAOM. L45-49.

Rationale for “water content” treatments need to be introduced before the Results

We added an explanation for why soil water content was added as a treatment in L85-87. We wrote “We incubated...at the lower and higher ranges of water filled pore space under non-drought conditions, which is a strong abiotic control on microbial activity.”

L106: Fig S6 shows that DOC was higher in the Ca treatment at high water content at the beginning of the experiment. These interactions are important and challenge the simple interpretation of main effects.

Following your suggestion, we clarified our point L300-302.

Briefly, yes, visually there seems to be a difference but it is not significantly different at $\alpha = 0.05$ ($P = 0.068$). We state the lack of difference in the figure (Now Supplementary Fig. 5) caption but do not include the full statistical analysis for DOC and MBC to avoid unnecessary supplementary tables (we already have many tables and figures in the SI). Even if the difference is significant, we do not think it changes our conclusions, as we explain below.

With respect to lower CO_2 following Ca treatment, we claim that it was not due to lower availability of DOC (because DOC was not different or tended to be higher; whether DOC is even a good measure of bioavailability in the first place, is debatable), since DOC was never lower under Ca treatment (and perhaps higher at high water content as you stated). Also, if DOC production is in some way related to microbial decomposition of litter, then this specific data can actually be interpreted as greater litter processing following Ca treatment, which is consistent with C NEXAFS (Figure 4) and incorporation of litter into MBC calculated from isotope tracing (Fig 2D). It does not support the counter claim that Ca exclusively stabilizes C by increasing adsorption of DOC (which it probably also does to some extent), as that would have resulted in lower DOC compared to control (which we did not observe, as DOC was, again not significantly different, and tended to be higher with Ca additions).

L138: “The effects of Ca on microbial community structure and utilization of litter affected bulk SOC composition” your data don’t conclusively support the causal nature of this statement

We agree and rephrased the sentence. L162-163.

L147: what kinds of clay minerals, or are you referring to total clay-sized particles? This is difficult to interpret. I am not certain what this metric is telling us.

We are referring to clay minerals in the mineralogical context not the particle size context. We revised to improve clarity (L172-174, L458-461).

Briefly: The FTIR peak at 3620 cm^{-1} is attributed to stretching of structural OH in clay minerals (2:1 and 1:1 aluminosilicates), following the approach of Lehmann et al (2007, Biogeochemistry, ref #72) and Hernandez-Soriano et al (2018, ES&T, ref #73), even though other non-aluminosilicate minerals (allophanes, gibbsite) also have similar absorbance bands in this area of the spectrum (Parikh et al 2014, Advances in Agronomy, Vol. 126).

L161: What about ^{13}C ? Since the litter was ^{13}C labelled, this would seem to be another crucial element of the argument

Please see our response to major comment #2 above.

L168: I don’t understand precisely what you mean by “co-localization of Ca with organic matter regions”, how can this be quantified as a percentage? A little more description needed;

We quantified the co-localization using a segmentation which is based on a machine-learning algorithm into OM and mineral dominated regions and independently Al-dominated, Fe-dominated and Ca-dominated areas based on the intensity of the ions for each selected element. Cross-wise combination of two independent segmentations is then quantified using the overlap. Detailed information on the segmentation and co-localization is provided in the Supplementary Information sections (NanoSIMS, *Image analysis*).

We now clarify this in L480-483:

"For specific co-localization types identified by a machine learning-based classification, we calculated the ^{15}N ratio ($^{12}\text{C}^{15}\text{N}^-:(^{12}\text{C}^{14}\text{N}^- + ^{12}\text{C}^{15}\text{N}^-)$) to determine ^{15}N enrichment, the normalized N:C ratio ($^{12}\text{C}^{14}\text{N}^-:(^{12}\text{C}^- + ^{12}\text{C}^{14}\text{N}^-)$), and the normalized Ca:N ratio ($^{40}\text{Ca}^+ + ^{44}\text{Ca}^+:(^{40}\text{Ca}^+ + ^{44}\text{Ca}^+ + ^{12}\text{C}^{14}\text{N}^- + ^{12}\text{C}^{15}\text{N}^-)$."

We also changed the color code in Figure 6DE to clearly indicate the signal not assigned to Al, Ca, and Fe, that is assigned a 'none' classification. We added an explanation to the 'none' classification in the supplementary information (NanoSIMS section, Image analysis). We wrote:

"The Si:(Al+Si) ratio of the 'none' segments in the mineral-regions was approximately five times higher than the Ca, Al, or Fe regions, indicating that they are quartz rich particles, while in the OM-dominated regions, there was no specific enrichment of any of the metals in the 'none' regions, indicating OM-rich segments. We calculated the relative pixel proportions of spatial co-localization types (excluding the 'none' segments which did not correlate with any metal) and the ion ratios to compare their composition (Figure 6D-H and Supplementary Fig. 14)."

...would seem to be heavily determined by the methods used for selecting regions of interest from the NanoSIMS data.

Techniques such as NanoSIMS bring the benefit of spatial information but lack the benefit of averaging over the entire sample. A complementary approach of bulk and spatially resolved techniques were used in this paper to minimize biases stemming from anomalous regions of interests. Also, in order to make the NanoSIMS analysis as representative as possible we collected a large number of images ($n=37$).

L172: Is "CN:C" a typo (do you mean N:C ratio? Or if not, more concrete explanation and rationale is needed for this metric)

This is not a typo. It is usual for NanoSIMS measurements to investigate the ion ratio $^{12}\text{C}^{14}\text{N}^-/^{12}\text{C}^-$ to (1) distinguish the resin from the OM when sections are analyzed, and (2) report it as a signature for the quality/composition of the OM. N is measured as the cluster ion CN ($^{12}\text{C}^{14}\text{N}^-$) according to its improved ionization. This approach was used by Schweizer et al., 2017 GCB (10.1111/gcb.14014) to calculate N:C ratio in soils. Following your suggestion, we have changed CN:C ratio to "normalized N:C ratio" (as used in Fig. 6G) and explained its use L480-483.

L178: what is "Ca:CN", is CN this sum of C and N, or something else? Also, very confusing in the figures

CN⁻ is a cluster ion evolved from the sputtering with Cs⁺. This metric therefore gives the ratio of Ca to organic N. To clarify, we have changed this to Ca:N in the text (L480-483), in Fig 6H, and in the Figure caption.

L219: This statement is apparently based on the ^{15}N NanoSIMS data, but in principle some of observed N associated with minerals could have been NH_4^+ mineralized from litter, or NO_3^- (could constrain this potential contribution using reasonable estimates of N mineralization based on stoichiometry). More importantly, if the argument holds, it could be bolstered by reporting the ^{13}C NanoSIMS data.

Please see our response above to your main comment #2.

L274: please provide further details of the agroecosystem management. What crop type, how was it managed, fertilization history, etc.

We have added additional details to Methods Section, *Soil sampling*, L352-355.

L278: the $\delta^{13}\text{C}$ is wrong (negative sign needed)

Thank you, corrected!

L319: I don't see why you need to make this assumption—since you are measuring CO_2 and $\delta^{13}\text{C}$, you can calculate litter decomposition directly. Or is this merely a heuristic for deciding when to end the incubation, based on the KOH data?

The latter. Since we measured $\text{d}^{13}\text{C}-\text{CO}_2$ on $\text{Ba}^{13}\text{CO}_3$ precipitates (Methods Section, *Experimental Design*), and due to logistical issues that isotope results were only available to use after the incubation was complete.

L347: I don't know what you mean by "six replicates were randomly paired"—what is being paired here, presumably each replicate is measured separately (if not, why not)

For isotope analysis we only measured 3 replicates (out of the 6 experimental replicates) due to cost. Instead of selecting 3 of the 6 microcosms, we paired them up to yield 3 replicates. We clarified this now in L397, L408-409, L426.

L435: I think you mean $p_{adj} > 0.05$ for non-significant ASVs?

Yes, corrected.

L437: I don't understand your linear models—are you modeling each ASV, or some multivariate index? Please provide more details

These methods were remnants from a prior analysis that was excluded from the present draft due to limited word count of the *Nat. Comms.* format. This sentence has been removed.

Fig 1B: The $P < 0.001$ appears to refer to the test comparing the low/high water content, but this is not of interest for the main question (Ca effect on litter decomposition); please clarify if there were differences between the Ca/control at each water content

There were no differences in fraction of CO_2 from the litter between Ca/control (L104-106). We removed the P value.

Fig. 1E,F: In my mind, these panels are misleading because they obscure the importance of the time x water interaction that you show elsewhere.

Following your suggestion, we revised Figure 1 (now Fig 2) – we now show the results obtained for low and high water content at T_{beg} and T_{end} for each variable. Additional data (fractionation results, DOC, MBC) which were less important to the main storyline are presented in the SI section Supplementary Figs 5-9. Statistics is provided in Supplementary Tables 2-5.

Fig 3: In my opinion, the Ca vs control spectra should be plotted on top of each other, with different time points shown in panels. We care more about the Ca effect than the time effect, and it is impossible to discern the Ca effect with the current layout

We replotted the C NEXAFS data according to your suggestion. It now more clearly shows an observable difference in O-alkyl relative abundance at T_{end} between the treatments. Upon further scrutiny we removed the statement describing differences in aromatic and substituted aromatic C. We revised the text accordingly (L166-168).

Fig 4: In my view the statistics are not appropriate for the question at hand. What we care about is whether the slopes of these regressions significantly differ between the Ca and control treatments, not their absolute values. I suspect that there may not be statistical evidence for different slopes in several of these panels.

Please see our response to your comment #3 above.

Also, the text in panels b,c,d seems incorrect (check the colors)

Thank you! Corrected!

Fig 5: I think there is a mistake, there is no Ca:CN shown

Thank you! We corrected some typos/errors in the figure legend.

692: I don't understand the rationale for plotting $(^{12}\text{C}^{14}\text{N}) : (^{12}\text{C} + ^{12}\text{C}^{14}\text{N})$, this needs to be clearly articulated

Please see our response to your similar comment above regarding L172. This ratio is calculated to determine the normalized N:C ratio. We modified the text to clarify (L480-483).

Fig. 6: what is the experimental evidence for part e. of the diagram? How can it be argued that the cycle intensifies over time? If this is actually correct, then I would assume that the new surface-colonizing bacteria would also be consuming some of the "MAOM", which might be consistent with the absence of observed increase in MAOM 13C in this experiment

Thank you for this comment! We agree that (re)cycling of newly formed MAOM by the surface colonizers is likely. Following the revision of Figure 2 to include results at T_{beg} and T_{end} , it can be seen that the fraction of MBC from litter-C is still significant, and significantly different in Ca-treated soils at T_{end} . Assuming the microbial population has

turned over several times in the 4-month duration, this means that microbes are still utilizing litter-C, a portion of which is likely MAOM-C, and perhaps again depositing it on mineral surfaces. We added this point to L263-267.

This does not contradict the idea of the intensifying cycle over time, for which the evidence is:

1) increased impact of Ca on bacterial community composition (Fig 3). The trends observed after 4 days (T_{beg}) where Ca and litter explained 28% and 32% of the variance in community composition respectively, were stronger after ~4 months (T_{end}) when Ca and litter explained 32% and 18% of the variance, respectively. That is, the impact of Ca on bacterial community selection intensified over time.

2) increased impact of Ca on co-localization of clay minerals and organic functionalities as shown by FTIR microscopy (Fig 5 and Supplementary Table 8). The increase in regression coefficient and R^2 values from T_{beg} to T_{end} was higher for Ca than control, suggesting that the impact of Ca on organo-mineral associations intensified over time.

3) increased impact of Ca on litter processing as shown by a decrease in relative abundance of O-alkyl in C NEXAFS (Fig 4 and Supplementary Table 6).

Fig. 7 should be placed first, as Fig 1, to introduce the reader to the experiment

In response to the referee's comment, we changed the order of the graphs. Fig 7 is now Fig 1, Fig 1 is Fig 2, etc.

Table S2: clarify the contrast control-Ca; does a negative t ratio imply a greater value in the control, or Ca treatment?

See our response to your major comment #5 above. Briefly, we removed the t statistic and show the effect sizes differences using the Cohen's d estimate. A negative value implies a larger mean value under Ca treatment. The estimate takes the same unit as the variable tested.

Figure S5: It is misleading to draw boxplots for only three reps per treatment, please remove the boxplots and just show the points.

We made the change.

Response to Reviewer #2:

In this manuscript the authors completed microcosm experiments on a Mollic Endoaqualf silt loam soil collected from a fallow subplot in Ithaca. The authors added $^{44}\text{CaCl}_2 / \text{CaCl}_2$ and $^{13}\text{C}^{15}\text{N}$ labelled willow leaves to different microcosms, destructively sampling after 4 days and ~4 months. The authors measured changes in microbial biomass, litter incorporation, community composition, carbon use efficiency, and respiration rates. They also completed standard characterization, spectroscopic measurements (bulk C/Ca NEXAFS; FTIR spectromicroscopy), density fractionation, and NanoSIMS measurements on their microcosms. The authors demonstrate that Ca addition reduced respiration rates by a small amount relative to the control (5%), influenced microbial community composition, carbon use efficiency (40%), and increased incorporation of litter into microbial biomass by 20%. Through the NanoSIMS and Ca labelling work the authors also novelly demonstrate ^{44}Ca enrichment and increasing co-localization of Ca with organic matter (68-90%) upon Ca addition. This work is novel and furthers our understanding of the role of Ca in SOC dynamics, incorporating its influence on microbial communities.

1. I think that this manuscript could be published in Nature Communications once the authors address several key major revisions. In their discussion, the authors stick closely to a single narrative, but I believe they could instead explore a range of potential explanations of their data and be more cautious in their interpretations. Particularly when their dataset is in disagreement with other Ca incubation studies^{18,25}. A lot of the interpretations seem to hinge on the increased incorporation of litter derived C into the mineral associated organic matter pool, yet unless I'm mistaken, the percent of ^{13}C incorporated to the MAOM pool from litter was lower in the Ca addition sample (T_{beg}) before being equal (if not slightly less) (Fig. 1E). Furthermore, I don't feel as though the effects of the pre-incubation / leaching step or KCl control data are adequately covered in the discussion, and as such, will need to be incorporated more clearly. I would urge the authors to be cautionary in their interpretations and language, exploring all potential influences of their methods and

causes of their results in the discussion. I thank the authors for moving the field forward and hope that they will submit a revised manuscript to Nature Communications.

We thank reviewer #2 for his positive feedback and constructive comments and suggestions. Please find below our response. We broke down some of your comments into individual questions to better provide a point-by-point response to each individual comment.

We have addressed your two main concerns: the incorporation of ^{13}C into MAOM, and the preincubation stage, as well as all other issues raised below. We are confident that with the help of your feedback we have improved the manuscript.

Major comments

2. Please report and discuss all T_{beg} and T_{end} . Results. While Ca addition affected CUE and ^{13}C transferred from litter to microbial biomass at T_{beg} , it had no effect at T_{end} ? This data should be displayed in the main graph and this result should be more clearly discussed. The results at T_{beg} are discussed in some locations and T_{end} in others. If we are to infer any conclusions about the short to medium-term effects of Ca addition on SOC dynamics, both observations should be reported and explained clearly (Fig. 1 in particular).

Following the suggestions of both reviewers we have revised Figure 1 (now **Fig. 2**) which now includes the data for both time points. We thoroughly revised the first section in the results, entitled "Calcium alters microbial respiration, litter metabolism, and conversion to MAOM," to improve comparison among timepoints. The rest of the figures already include both time points, except NanoSIMS for which only T_{end} samples were analyzed.

Specifically: the fraction of MBC derived from litter-C is significantly higher following Ca addition at T_{beg} and at T_{end} .

3. Please report the effects of Ca addition on the MAOM fraction clearly. Line 107 (also line 180-183, 219-221) states that Ca influenced the transfer of litter into the MAOM fraction, which to the reader, implies that Ca probably increased the incorporation of ^{13}C from litter into the MAOM fraction. Yet, in your results it actually seems that at T_{beg} (Fig. 1E) there was less C being incorporated to MAOM in the Ca addition samples, seemingly Ca addition reduced MAOM incorporation until T_{end} when there was no difference. This needs to be made far clearer in this section and throughout the manuscript, see some other locations above.

Thank you for highlighting this issue. We have clarified these points in the Discussion (L261-270).

"In the Ca-treated soils, a larger fraction of MBC at T_{beg} was derived from litter-C than in control soils, indicating that more litter was cycled through microbial biomass, while concomitantly less MAOM from litter was measured in Ca-treated soils. This is likely an observation of a transient nature at T_{beg} which captures the greater uptake of litter-C into MBC prior to its conversion into MAOM. At T_{end} , we found that the amount of litter-derived C and N transferred to MAOM was not different (Fig. 2E), which may be explained by the fact that surface-adhering bacteria likely also consumed litter-derived MAOM throughout the incubation. The observation that the proportion of labeled C in MBC was significantly higher after Ca addition even at T_{end} (Fig. 2D and Supplementary Table 2) supports this claim. So does the observation at T_{end} that MAOM consisted of compounds that were more microbially processed (lower C:N) after Ca addition (Fig. 2F). Thus, MAOM composition, but not amount, was impacted by Ca addition."

Confirming the role of microbial processing in conversion of plant C into MAOM C is the observation that at T_{end} , after 4 months without any additional exogenous C input, MAOM- ^{13}C was higher at high water content. We believe this is due to the greater microbial activity at higher water content, which increased conversion of litter-C to MAOM-C (Figure 2E).

We also propose explanations for why isotope/elemental analysis did not find differences in MAOM between Ca and control, while FTIR microscopy did show differences (L283-288). We wrote:

"The discrepancy between FTIR-microscopy analysis which indicated greater abundance of organic functional groups in Ca-treated soils and elemental and isotope analysis that did not show differences (Fig. 1E, Supplementary Fig. 7) could stem from the spatially resolved nature of the former vs bulk characterization of the latter, and from the fact that FTIR-microscopy was done on microaggregate (53-250 μm) sections while the isotope analysis was done on $> 53 \mu\text{m}$ particles."

4. Following on from my last comment, according to line 158 your spectromicroscopy and line 168 NanoSIMS results indicate increased formation of organo-mineral association because of Ca addition, but this was not from the litter as per Fig. 1E.

Please see our response to your comment #3 regarding transfer of litter to MAOM.

We agree that the spectromicroscopy techniques and elemental and isotope analyses do not fully agree on whether more litter C&N was transferred to MAOM in Ca treatment, or whether just the composition of that OM and its spatial distribution changed. This is likely to some extent a result of the different sensitivities of the methods and the bulk vs. spatially resolved measurements that they obtain. We added this to the discussion as shown in our reply to comment #3. Please see our reply to the next comment regarding changes to MAOM-C

Please report all of your density fractionation data. How did your density fractionation results change? Why aren't these reported? Maybe I missed them, but I couldn't find them in your results or SI. You mention on Line 346 that six replicates were randomly paired and measured, but I don't see the results from the sample mass, C, N, $\delta^{13}\text{C}$ or $\delta^{15}\text{N}$ of each fraction? This seems like it would be an easy way to quantify an increase in mineral associated SOC caused by Ca addition?

Thank you for these suggestions. We added all the fractionation results to the SI.

fPOM, oPOM, and MAOM mass distribution (mg/g soil) for all treatments and time points is now provided in Supplementary Fig. 6. Please note that there are five to six replicates for mass. No difference in fraction amounts between treatments was found.

Supplementary Figs. 7,8, and 9 show soil and litter derived C and N (mg C or mg N/g soil), in fPOM, oPOM, and MAOM for both treatments and time points. These values integrate atom%, C or N content, and relative amount of fraction in the soil and are the best representation of litter and soil derived C and N pools. Please note that this data is replicated in three times.

There are no significant differences that would suggest an increase in MAOM-C by Ca addition. See our response to your comment #3 regarding this. Rather, MAOM composition (C:N) and spatial distribution changed.

Your recoveries also seemed high for your density fractionation (109%), was the SPT adequately washed out from these fractions? Did you measure the EC of your rinse solution?

Each fPOM and oPOM sample was washed on the filter with 500 mL DW. The EC was measured but not recorded for every sample, but for those that were checked it was similar to the DW, and we therefore concluded that the SPT was adequately washed out. The remaining pellet (MAOM+sand) was washed until the supernatant density (a measure of SPT residue) was $1 \pm 0.02 \text{ g/cm}^3$, which indicates no SPT (1.65 g/cm^3) residue in the fractions.

Mean mass recovery was 102% (L433). This may have resulted from the sodium-hexametaphosphate used to redisperse the pellet. The Na-HMP cannot be removed since the $<53 \text{ um}$ is dried along with it. However, this should not change any of our findings.

The C recovery ranged from 96-109% and N from 98-106%. This likely reflects the sum of all variance and errors in the fractionation process and sample analyses.

5. How sure are you that the addition of Ca selected for these microbial taxa because of their survival strategy as surface adhering organisms, and thus, their ability to excrete substances, such as extra-cellular polysaccharides, for which Ca is a critical co-factor? How sure are you that these taxa were only selected for because of their supposed ability to adhere to surfaces and not, for instance, their ability to tolerate stress, be versatile, or have a range of functional traits that benefit from Ca addition?

We agree with the need for cautious interpretation. Bacterial 16S rRNA phylogenetic gene marker data cannot offer definitive proof regarding the forces driving the shifts in microbiome we observed. We now acknowledge this in the Discussion, writing: "The abiotic and biotic factors that fostered the growth of surface-colonizing populations cannot be discerned from phylogenetic gene marker data and are likely a result from a combination of factors (see Supplementary Information for more details)." (L245-248). Still, we draw a degree of confidence from several lines of

evidence that Ca treatment selected for populations adapted to surface colonization rather than merely stress tolerance, and we note that these physiological traits (as well as others) are not mutually exclusive.

For context, stress tolerance in bacteria manifests in diverse ways and are not easily discriminated by phylogenetic gene marker data, whereas the surface-colonizing groups that we observed all have specialized growth habits for surface colonization (hyphae), surface gliding motility, or dimorphic surface-adherent lifestyles, which are all phylogenetically conserved traits. Thus, while we have confidence that our phylogenetic groups possess traits of surface-colonization, we cannot definitively say their enrichment in Ca treated soil is or is not related to stress tolerance. However, because the exposure to salinity stress during the pre-incubation was transient, we considered the temporal response of bacterial groups as an indicator of the potential impact of salinity stress. We found that most of the major surface-colonizing groups responding to Ca treatment (*ex. taxa* from the *Hyphomicrobiales* and *Xanthomonadales*) had the same relative abundance in control and Ca-treated soils at T_{beg} but diverged in Ca treated soils over time. This was captured by the broad trend in the increased amount of variation in community composition attributed to Ca treatment over time (Figure 3AB). We now draw attention to this, writing: "...a few major groups were favored by Ca (orders *Hyphomicrobiales* and *Xanthomonadales*) and these populations had increased relative abundance compared to controls only at T_{end} ." (L153-155)

The one exception to the temporal trend in surface-colonizing populations was the hyphae-forming actinobacterial groups, which were most differentially enriched by Ca at T_{beg} , and less so after 4 months (Supplementary Figure 11; also provided below). These actinobacterial groups are recognized for their tolerance to osmotic stress, and it is plausible that exposure to high salinity during pre-incubation may have factored in their early high differential abundance. However, this does not preclude the importance of their growth habit. For example, we know that hyphae-forming *Actinomycetota* are favored in soils when pores are less hydraulically connected due to their filamentous growth (Wolf et al., 2013, ref #15 in SI). We found evidence of this phenomenon when comparing the relative abundance of the same actinobacterial groups between high and low moisture conditions. It is important to note that our low moisture condition (40% WFPS) is within the range of normal / non-drought conditions in well-drained soils and does not represent a high stress condition. Thus, the selection for these actinobacterial groups in low moisture soils offer evidence that their growth habit was a factor in their predominance. We conclude that the most likely scenario is that the diverse surface-colonizing bacterial populations were enriched by Ca due to changes in soil properties related to Ca and OM, and other factors, including potential stress. We have now provided the caveat on L244-257 and discussed this more thoroughly in the Supplementary Information. Furthermore, we highlight in the Discussion (L237-240) that many of the same taxa were favored in a study which amended calcitic lime to forest soil, underscoring a degree of consistency among Ca-containing amendments.

Supplementary Figure 1. Ca-treatment produced an increase in relative abundance of major bacterial groups relative to control soils, including members of the (A) *Actinomycetota*, (B) *Hyphomicrobiales*, and (C) *Xanthomondales*. Notably, time-dependent differences in the effects of Ca were observed, with the increase in *Actinomycetota* most pronounced at T_{beg} , and subsiding at T_{end} , while the enrichment of *Hyphomicrobiales* and *Xanthomondales* by Ca-treatment was only apparent at T_{end} . The plot shows the aggregated relative abundance of several genera for each Phylum or Class. The significance of the main effect of calcium is displayed in each panel, according to results of linear regression models. Lettering denotes significant differences in mean DNA yield among litter and moisture treatment according to TukeyHSD ($p_{\text{adj}} < 0.05$).

Furthermore, I believe that actinobacteria which form mycelia also coat themselves in Ca oxalate as a strategy. Could you rule out other explanations for your data? If it was purely due to the increased excretion of surface-adhering compounds, then why do you see a decrease in your O-alkyl C content in the Ca addition treatment? Wouldn't this region of C K-edge spectra (289.4 eV) increase if there was more abundant excretion of these compounds by organisms adhering to surfaces? If Ca was instead stressing the microorganisms or causing them to succeed due to other strategies, this could also explain your observations? This needs to be addressed properly in the discussion. I would urge the authors to be more cautious in their interpretations and present all possible explanations of these changes in microbial community composition.

Bulk C NEXAFS is likely not sensitive enough to detect potential changes in the composition of compounds exuded by microbes under different conditions. It is a more likely explanation that the decrease in the signal at 289.4 eV was a result of litter decomposition. We think that it is plausible that the byproducts of surface-colonizing populations may

affect composition, including the materials promoting adherence to surfaces, but we do not argue for a specific origin in our article.

We now discuss this in L270-275:

"Additional evidence of compositional changes in SOC upon Ca addition comes from bulk C NEXAFS analysis which shows lower relative O-alkyl abundance (289.4 eV) at T_{end} than in the control soils (Figure 4B). While this could have theoretically resulted from a shift in the composition of compounds produced by microbes after Ca addition, the most reasonable explanation is that litter decomposition was driving O-alkyl reduction, as previously reported for incubation studies⁴⁹."

Lastly, we could not find evidence in the literature to support that *Actinomycetota* secrete Ca oxalate. However, *Actinomycetota* are known for the oxalotrophic activity (consumption of Ca oxalate), yielding the biomineralization of calcite, though the process is not specific to Actinobacteria.

6. It seems that the KCl experiments are extremely important as a control? Why were they not run for the same extended period of time? Furthermore, what was the effect of KCl addition on the microbial community composition? Is it possible that the changes that you have recorded could just be an effect of increasing ionic strength on the microbial community? Please report all available data on the KCl control and discuss it clearly in the discussion.

The KCl experiment was conducted to evaluate if there was an ionic strength effect on mineralization. While it may have been valuable to have mineralization data for KCl-treated soil throughout the duration of the incubation period, the differences between KCl and CaCl_2 were evident from the very beginning of the incubation. In our experience, differences across soils or treatments are apparent at the early stages of incubations; the rest of the experiment usually follows the same trajectory, at least with regards to CO_2 . While it is possible that the punctuated rise in salinity during preincubation affected the community's ability to mineralize C, our observations do not align with the expected impact of salinity, which should lower CUE, as physiological stress increases maintenance costs that reduce efficiency (Manzoni et al., 2012, *New Phytologist* doi.org/10.1111/j.1469-8137.2012.04225.x; Malik et al., 2018, doi.org/10.1038/s41467-018-05980-1). Furthermore, the impact of pulse disturbances on microbial community structure and function tends to be greatest in the immediate aftermath, with communities converging in similarity to reference communities over time (Hillebrand and Kunze, 2020, *Ecology Letters*, doi.org/10.1111/ele.13457). We observed the differences in community composition between control and Ca treated soils to increase, rather than diminish, over time. Thus, while we cannot rule out the effect of salinity, discussed on L305-313, we are confident that our set of observations provide evidence for the specific role of Ca.

Briefly: Evidence for the role of Ca includes the changes in physical-chemical characteristics of OM, which selected for surface-colonizing bacterial taxa, which were also similar to changes in the microbiome observed in limed forest soils (Sridhar et al., 2022 ref #27 in manuscript– see section entitled: "The effects of calcium from lime").

We did not collect 16S rRNA gene sequences or perform isotopic analyses on the KCl soils due to budget and labor limitations.

7. Line 232 Do you believe that your DOC measurements could have also been affected by your pre-treatment? It seems that you measured the DOC in your MBC analysis and not in the leachate of your pre-incubation step. Yet, you also pre-incubated and then leached your soils? What would be the effect of this on DOC content? What if the DOC (that had not been bound to Ca in your control) had already been flushed from your samples during this step? Was this measured? Are you sure that DOC export was not reduced like has been seen in other studies¹⁸, particularly at circumneutral pH²⁵, like your own study?

We greatly appreciate your thoughtful and cautious approach weighing our observations. Indeed, a reduction in a C pool size (e.g. DOC) resulting from the preincubation could have reduced CO_2 emitted. While we did not measure the DOC of the leachate, the total organic C after preincubation was the same for control and Ca (**now added to Supplementary Table 1**).

It is possible that 'non-bound' DOC in the control was removed to a greater extent than in the Ca-treated soils. Indeed, DOC after the preincubation (supplementary Table 1) was higher in the Ca-treated soil (added to Supplementary Table 1.) Therefore, if DOC constitutes a more labile C source that significantly contributes to soil respiration, we would have expected higher CO₂ in the Ca soils, but we observed the opposite. Since DOC in Ca-treated soil was not significantly different than control at Tbeg (Supplementary Fig. 6) it was therefore not the driver of lower CO₂ emission. We discuss this in L298-305 and point out the differences between our results and those in the literature and the very different soil properties in these studies.

Why was this pre-incubation step taken? and please discuss it more clearly in your discussion.

We appreciate this advice and have provided more details about the rationale for our approach in the methods section (L370-373). It is a challenge to change soil Ca content without unintended changes to pH or ionic strength. Our approach was a hybrid of the methods used by Marinos and Bernhardt (2018), who directly added CaCl₂ to the soils but depended on plant watering to leach excess salts (soil solution EC wasn't reported in that study), and Whittinghill and Hobbie (2012), who performed an elaborate cation exchange protocol followed by washing steps to remove excess salts, but at the expense of destroying the soil structure.

8. Line 247 This is an interesting conclusion because in Supplementary Table 1, the main changes to your exchange complex are a reduction in Mg, another divalent cation. The monovalent cations Na and K barely change between your treatments. Thus, if Ca is replacing Mg on the exchange complex, why is this creating new sites for microbial attachment? Wouldn't you expect these sites to be somewhat similar due to the divalency of both cations? Please explore why this is the case in your discussion and suggest all possible explanations of your data, being more cautious in your interpretations.

We added this discussion point in L321-323, L326-329. While both are divalent it has been shown that Ca and Mg are dissimilar in terms of surface interaction (ref #58), and in the context of inducing surface colonizing behaviors (ref #21). Thus, the functions of Mg and Ca may not be identical with regards to SOC stabilization.

We wrote:

"Despite both Mg²⁺ and Ca²⁺ being divalent cations, they likely function differently. Indeed, organic matter affinity to Ca²⁺ was found to be greater than to Mg²⁺ (ref 60). Ca²⁺ is also required for inducing surface- colonizing such gliding motility, and cannot be replaced by Mg²⁺ (ref 21)."

Minor comments:

Line 22 Replace increase with contributes to.

Replaced

Line 29-31 The meaning of this sentence is convoluted by its length. I'd suggest breaking up the arguments.

Done

Line 36 Exchangeable Ca can also be found on the exchange sites of organic matter and other negatively charged soil minerals, not solely on clay minerals.

We agree, and therefore said "**mostly** found on clay minerals". We are concerned that it would be confusing to have the first sentence state:

"Globally, there is a positive correlation between exchangeable calcium (Ca), mostly found on clay minerals and soil organic matter, and soil organic carbon (SOC) content in slightly acidic to alkaline soils".

When corrected for the exchangeable Ca contributed by SOM, the relationship between Ca(ex) and SOC content still holds, and apart from SOM, clay minerals are usually the biggest contributors to CEC. We have decided not to change this sentence.

Line 46 and physically protected within the soil matrix?

We agree. However, the types of associations are already listed in paragraph 1 L41-43 *"...by driving physico-chemical associations between organic compounds and minerals, such as sorption, co-precipitation and complexation, and occlusion within aggregates"*. We do not think it is necessary to restate this.

Line 49 soil mineralogy also?

Yes, we agree. Added.

Line 74-75 Shouldn't this be a separate point?

Yes, we broke up Q3 into two different questions (L79-80)

Line 83 I don't believe that bulk Near Edge XAFS requires capitalization in its full form here.

Change made.

Line 166 There is an extra unbalanced bracket here.

Thank you. Corrected.

Line 166-167 This sentence probably needs to come before Line 164 as it sets up the preceding argument.

We agree. Change made.

Line 168-170 How precise are these percentages? Depending on their precision, I'd suggest reporting these percentages to the nearest whole number?

Changed to nearest whole number.

Line 195 – 198 Overall Ca addition selected against the communities that were positively enhanced by litter addition and reduced the incorporation of this litter to MAOM at T. Beg.?

By highlighting changes in populations associated with litter amendment, we provide evidence that Ca impacted the specific populations necessary to influence microbial processing of litter C. Had we not observed any such differences, we would not have clear evidence that changes in microbial activity could explain observed differences in C processing. However, while our evidence shows the necessary preconditions are met (i.e., altered decomposer populations) we cannot draw conclusions about whether changes should increase or decrease respiration from our phylogenetic gene marker data. One could arrive at plausible explanations for why a reduction in the relative abundance of decomposers populations could produce either outcome. Given other observations, the most logical explanation would be that Ca selected for a subset of bacteria (small proportion of DNA pool), whose exhibited higher levels and/or more efficient levels of activity, yielding a reduced respiration, but higher incorporation into MAOM. In discussing these results, we simply state that the reduction in relative abundance corresponded with lower respiration and higher CUE, writing: "This observation suggests that Ca treatment restructured decomposer communities, and that the reduced net respiration and increased CUE resulted to some extent from a change in decomposer populations. However, we cannot test this relationship directly, as we lack information about changes in the absolute abundance and activity of these populations." (L245-248)

Line 225 Rowley et al., 2021 <https://doi.org/10.1007/s10533-021-00779-7> specifically supports your density fractionation observations regarding the importance of the mineral-associated pool and reduced importance of the occluded fraction as per line 111 to 118.

We agree and added this reference (L283).

Line 226 Does this not also show that Ca is contributing to the preservation of plant-like organic compounds that are rich in aromatic and carboxylic moieties?

Differentiation of microbial vs plant C is an important and ongoing debate (e.g., Whalen et al 2022, GCB, 10.1111/gcb.16413; Angst et al 2021 SBB) and we agree that a nuanced and cautious approach is warranted. We edited this statement to reflect this L293-295.

We also added that the FTIR peaks assigned as "aromatic-C" also consist of other functional groups not necessarily derived from plant including N-H bending and C=N stretching (~1545 cm⁻¹), and C=O stretching (~1650 cm⁻¹) (L460-462).

Line 232 This is contrary to the results of Minick et al., 2017 <https://doi.org/10.1007/s10533-017-0307-z> your reference 18 and should be discussed more clearly. As you mention in your intro, Minick saw a reduction in the mineralization of SOC and enhanced mineralization of litter. This could also be covered more clearly in your

results. What do you think the effect of your pre-incubation could have been? Could this have also influenced your DOC results?

We added a comparison to other papers in L302-205. Please see our response to your comment #7 above. Indeed, we found different results than Minick et al., 2017. However, those soils had much lower initial pH, the wollastonite treatment increased pH as well as Ca, and third, the cation exchange capacity of the soil in that study was much lower.

Please see our response to the comment on preincubation above.

Line 253 Doesn't the higher proportion of aromatic C moieties indicate that this C is slightly less processed than your control soil?

1. Please see our response to your comment on L226 above.
2. We also base our statement on lower C:N values (Fig 2 isotopic analysis and Fig 5 NanoSIMS analysis).

At current it is not clear how many replicates were made in the different treatments? Was it 6 for each treatment? I believe so, but it is not very clear as currently worded in the methods. Including the number of observations throughout, even the figure captions, would be useful for the reader and more clearly delineating this in the methods section will also be important.

We edited the appropriate sections in the methods section to clarify this (L397, L408-409, L426, L434-435) and in the figure captions.

Briefly: six jars were prepared for each treatment and duplicated for Tbeg and Tend. Three of six jars were used for ¹³CO₂ and CO₂ measurements; all six jars were used for density fraction mass determination; for elemental and isotopic analyses (microbial biomass, DOC, soil fractions) the six jars were paired randomly and combined to yield three replicates.

What was the influence of your pre-incubation step? Why was this taken? Did you measure changes in microbial community at this stage also before leaching your soils, as part of a disturbance control? Did you also measure the effects of KCl on microbial community composition? Please report the KCl data more clearly. Please see our responses to the comments above regarding these questions.

Figures:

Please present T End also in Fig. 1. or consider breaking this up into multiple figures. Consider moving the explanation of the error bars for A up to the correct location (I was about to enquire about what the error bars meant).

We revised Fig. 1 (**now Fig. 2**) to include all time points and both water contents. We moved the explanation.

Supplementary Fig 2, 3. While the Ca L-edge has not shifted, I'm not sure that this lends credence to the idea that Ca addition did not affect Ca speciation. I think this would be hard to see at the Ca L-edge, which is less detailed than at the Ca K-edge. I'd suggest being more careful with your interpretations here. Thank you for this comment. We agree that Ca L-edge NEXAFS can provide only limited information. We revised the relevant text in the SI to reflect that.

Supplementary Fig 2, 3. Is there also a normalization issue with some of your standards (Ca L-edge) particularly the calcite, citrate, and formate?

Yes, there is a little dip at ~347 eV. We are not sure why.

Tables:

Please add all available data, including the density fractionation work.

Added. See response to comments above.

Please add a table detailing the effects of the pre-incubation step on your soils, prior to the start of your experiments.

This was provided in the original submission in Supplementary Table 1. The Table details all the analyses done on the soil following the Ca or control treatment, including pH, EC, exchangeable cations, cation exchange capacity, mean weigh diameter after slow and fast wetting, and aggregate slaking value. We have added our measurements of DOC and total C in preincubated soils to Supplementary Table 1.

Reviewer #1 (Remarks to the Author):

I thank the authors for their thorough and thoughtful revisions, which helped me better understand and appreciate the arguments made in this paper. This is an impressive body of work.

I believe I found one small error in the revision: L107 states that KCl treated soils had lower C mineralization than Ca-treated soils, but Supplementary Fig 1 shows the opposite. Greater mineralization following KCl treatment supports the argument that the authors are making.

Reviewer #2 (Remarks to the Author):

Dear Dr. Shabtai et al.,

I commend you on your swift revisions. As suspected, I thoroughly enjoyed re-reading this article. I found that the introduction, abstract, and results section all read great. It also felt as though your figures more clearly depicted some of the uncertainty in the data, while highlighting the effects of Ca addition on your system. I am still worried by the DF results, but will discuss that further below. I look forward to the finalized published version of this manuscript and thank the authors for considering the comments from both reviewers.

I've suggested a few minor comments below and one more (semi-)major comment.

My semi-major comment is that I would be more skeptical in interpreting the results from your DF. You didn't see a statistical increase in your MAOM fraction (C or N) at t-end, as one would expect from your current conceptual model. Do you think it is possible that the DF could have destabilized some of the organic matter freshly bound in the MAOM fraction by your surface colonizers? When you think about the small increase in exchangeable Ca that you created through your addition, maybe an exchange with Na⁺ (over the 18 h and then extracted with SHMP), in turn influenced the fractionation results from your incubations? The inclusion of DF is supported by many other studies in Ca rich soils, but a more skeptical discussion on why MAOM didn't increase during your incubations is still required.

Thanks again and I look forward to eventually seeing this published.

Minor comments:

Line 95-97 - I'd suggest rewording this last sentence of the intro to make it punchier.

For the first time our results demonstrate mechanistically that Ca drives changes in the soil microbiome...

106-107 - I think the KCl treatment produced higher mineralization than the Ca-treated soils.

213 - I believe you are stating that it formed new organo-mineral associations. Potentially doesn't really work in its current form. However, referring to my semi-major comment, did it?

323 - Do you think that the exchange of old Ca may have destabilized the DOC that was bound to it and thus caused your higher DOC flush from your pre-incubation (Suppl. Table 1)? This might be the place to make this explicit if you feel it is accurate. It's an interesting result and worth discussing. It would also link into an explanation of your DF results?

Furthermore, this was a lot more DOC extracted from the Ca amended soil. From a quick back of the envelope calculation, it's several orders of magnitude off the response that you saw in CO₂-C respiration at a bulk level. However, it might be worth mentioning this.

357 - how was soil texture measured?

363 - were total element concentrations also measured? Total Ca etc? bulk mineralogy ?

369 There's a hyphen in-between 50 and mL and a space between 20 and °C that can be removed. There's normally not a gap between the degree sign and unit, unlike percentage and other forms. I would be careful to normalize your units throughout. Sometimes it's / L and other times -1.

436 Building on the last comment and passing on a constructive comment from a previous reviewer, delta is typically italicized throughout - $\delta^{13}\text{C}$ values.

Figures:

The figures are much clearer now, it's great that some of the data is more accurately represented. Especially in the C NEXAFS, you can really see that most of the change is in the O-Alkyl C region.

Conceptual model – It would probably be better to remove K as this did not change in your system. I think another Mg would probably be more relevant. There's also another reference in Kalinichev and Kirkpatrick (2007) that used molecular dynamics simulations to investigate the different behavior of organic functional groups in the presence of Ca and Mg.

The suppl. figures for the DF results have errors in the y legends.

Ca L-edge data - It does seem that there is a real issue with the normalization of this data. Could you please attempt to renormalize it? As mentioned in my previous comment, I'm also not sure that you would have the spectral resolution at the L-edge to effectively see if you had changed the Ca-speciation of your soil. I believe you'd see more of a shift if you had limed the soil or applied Ca silicates, relative to increasing the CaExch concentration by a small fraction. I would suggest reducing the emphasis on this towards 'we did not detect any changes in Ca speciation at the L-edge', just to be conservative. Furthermore, it seems that you would require a good normalization of both the control and Ca treatment samples if you are to compare the splitting ratio between them. The splitting ratio has only been conservatively used as an index of crystallinity with critiques of this posed by Cosmidis et al. (2015). I think you could adjust your normalization parameters slightly in Athena and be able to report this data more accurately, even accounting for the bump.

SI methods:

Please include the normalization parameters in Athena, where you placed E0, the pre and post edge bounds, etc.. for both the Ca and C edges on your bulk NEXAFS SI.

Cosmidis, J., Benzerara, K., Nassif, N., Tyliszczak, T., Bourdelle, F., 2015. Characterization of Ca-phosphate biological materials by scanning transmission X-ray microscopy (STXM) at the Ca L_{2,3}-, P L_{2,3}- and C K-edges. *Acta Biomater* 12, 260-269.

Kalinichev, A.G., Kirkpatrick, R.J., 2007. Molecular dynamics simulation of cationic complexation with natural organic matter. *European Journal of Soil Science* 58(4), 909-917.

VIEWER COMMENTS

Reviewer #1 (Remarks to the Author):

I thank the authors for their thorough and thoughtful revisions, which helped me better understand and appreciate the arguments made in this paper. This is an impressive body of work.

We appreciate your positive remarks.

I believe I found one small error in the revision: L107 states that KCl treated soils had lower C mineralization than Ca-treated soils, but Supplementary Fig 1 shows the opposite. Greater mineralization following KCl treatment supports the argument that the authors are making.

Corrected!

Reviewer #2 (Remarks to the Author):

Dear Dr. Shabtai et al.,

I commend you on your swift revisions. As suspected, I thoroughly enjoyed re-reading this article. I found that the introduction, abstract, and results section all read great. It also felt as though your figures more clearly depicted some of the uncertainty in the data, while highlighting the effects of Ca addition on your system. I am still worried by the DF results, but will discuss that further below. I look forward to the finalized published version of this manuscript and thank the authors for considering the comments from both reviewers.

We appreciate your positive remarks. Please find below our point-by-point response to your comments.

I've suggested a few minor comments below and one more (semi-)major comment.

My semi-major comment is that I would be more skeptical in interpreting the results from your DF. You didn't see a statistical increase in your MAOM fraction (C or N) at t-end, as one would expect from your current conceptual model. Do you think it is possible that the DF could have destabilized some of the organic matter freshly bound in the MAOM fraction by your surface colonizers? When you think about the small increase in exchangeable Ca that you created through your addition, maybe an exchange with Na⁺ (over the 18 h and then extracted with SHMP), in turn influenced the fractionation results from your incubations? The inclusion of DF is supported by many other studies in Ca rich soils, but a more skeptical discussion on why MAOM didn't increase during your incubations is still required. Thanks again and I look forward to eventually seeing this published.

We agree that based on our conceptual model (and FTIR and NanoSIMS results) one would expect an increase in MAOM, and an artefact arising from the density fractionation might have affected those results. We now add this point to the discussion in L277-288.

"Additionally, we cannot rule out the possibility that the sodium polytungstate and sodium hexametaphosphate used in floatation and dispersion during the fractionation protocol may have exchanged Ca ions, potentially destabilizing some of the organic matter that was bound to the added Ca (refs #54,55). While Na-polytungstate is currently the best available option for density fractionation, a less harsh procedure for dispersal (e.g., shaking with glass beads) may be called for."

Minor comments:

Line 95-97 - I'd suggest rewording this last sentence of the intro to make it punchier.

For the first time our results demonstrate mechanistically that Ca drives changes in the soil microbiome...

Thank you for the suggestion. However, the Journal guidelines discourage the use of phrases like “for the first time”. We changed it to “*Our results show a hitherto overlooked mechanism by which Ca can change the soil microbiome and alter the conversion of plant-derived C into more persistent MAOM fractions.*” L91-93.

106-107 – I think the KCl treatment produced higher mineralization than the Ca-treated soils.
Corrected!

213 – I believe you are stating that it formed new organo-mineral associations. Potentially doesn't really work in its current form. However, referring to my semi-major comment, did it?
We believe that the spectromicroscopic results are robust enough to suggest the (reasonable) claim that Ca helped form new OMAs. We propose several possible explanations for the discrepancies between those analyses and the elemental/isotope analysis of the MAOM, and potential limitation of our fractionation method in L283-288.

323 – Do you think that the exchange of old Ca may have destabilized the DOC that was bound to it and thus caused your higher DOC flush from your pre-incubation (Suppl. Table 1)? This might be the place to make this explicit if you feel it is accurate. It's an interesting result and worth discussing. It would also link into an explanation of your DF results?

Firstly, the units for DOC shown in Suppl. Table 1 have been corrected to $\mu\text{g C g}^{-1}$ soil (not $\mu\text{g C L}^{-1}$).

Yes, the higher DOC shown in Suppl. Table 1 may have been associated with the Ca exchange and C bound to it, though we did leach **both** Ca and control soils to limit these types of effects on downstream measurements. We added this point to the methods (L370-372). We discuss the DF issue specifically in the discussion section (see comment above).

Furthermore, this was a lot more DOC extracted from the Ca amended soil. From a quick back of the envelope calculation, it's several orders of magnitude off the response that you saw in CO₂-C respiration at a bulk level. However, it might be worth mentioning this.

Referring to the difference shown In Suppl. Table 1: DOC difference of $\sim 70 \mu\text{g C/g}$ soil between treatments is equivalent to $\sim 1.8 \text{ mg CO}_2\text{-C/g SOC}$ (the mineralization units), which is similar to the difference in cumulative CO₂-C evolved at Tbeg (3 mg CO₂-C/g SOC). However, assuming that more DOC potentially drives more CO₂-C, we would have expected to see more CO₂-C in Ca-treated soils, but we observed the opposite trend – less CO₂-C evolved from Ca-treated soils. The CO₂-DOC relationship however is likely not that straightforward.

At Tbeg, 4 days after the extraction summarized in Suppl. Table 1, DOC concentration were not significantly different between Ca and control (visually higher at high water content but not statistically so). The higher DOC at Tbeg than Tend was likely derived from litter mineralization. Had the DOC of Ca soil been **lower** at this point, we would have been suspicious that this was related to the lower CO₂, again assuming the DOC-CO₂ relationship, i.e. our Ca-treatment and leaching impacted downstream measurements. However, since this was not the case, we are reasonably confident that any preferential DOC release during pre-treatment did not influence the results in a direction that refutes our main claim. We discussed this in the previous revisions in L289-293.

357 – how was soil texture measured?

Texture was determined using the hydrometer method following dispersal by Na-HMP (L348-349).

363 – were total element concentrations also measured? Total Ca etc? bulk mineralogy?

We did not.

369 There's a hyphen in-between 50 and mL and a space between 20 and °C that can be removed. There's normally not a gap between the degree sign and unit, unlike percentage and other forms. I would be careful to normalize your units throughout. Sometimes it's / L and other times -1.

Corrected. All units were changed to the form $^{-1}$, as per journal guidelines.

436 Building on the last comment and passing on a constructive comment from a previous reviewer, delta is typically italicized throughout - $\delta^{13}\text{C}$ values.

Corrected.

Figures:

The figures are much clearer now, it's great that some of the data is more accurately represented. Especially in the C NEXAFS, you can really see that most of the change is in the O-Alkyl C region.

Thank you. We agree that the representation of C NEXAFS data has improved.

Conceptual model – It would probably be better to remove K as this did not change in your system. I think another Mg would probably be more relevant.

We made the change.

There's also another reference in Kalinichev and Kirkpatrick (2007) that used molecular dynamics simulations to investigate the different behavior of organic functional groups in the presence of Ca and Mg.

Thank you for pointing out this useful reference! Inserted (ref #61).

The suppl. figures for the DF results have errors in the y legends.

Corrected.

Ca L-edge data - It does seem that there is a real issue with the normalization of this data. Could you please attempt to renormalize it? As mentioned in my previous comment, I'm also not sure that you would have the spectral resolution at the L-edge to effectively see if you had changed the Ca-speciation of your soil. I believe you'd see more of a shift if you had limed the soil or applied Ca silicates, relative to increasing the CaExch concentration by a small fraction. I would suggest reducing the emphasis on this towards 'we did not detect any changes in Ca speciation at the L-edge', just to be conservative.

We changed the text accordingly in the main text (L372-373) and SI Ca NEXAFS section.

Furthermore, it seems that you would require a good normalization of both the control and Ca treatment samples if you are to compare the splitting ratio between them. The splitting ratio has only been conservatively used as an index of crystallinity with critiques of this posed by Cosmidis et al. (2015). I think you could adjust your normalization parameters slightly in Athena and be able to report this data more accurately, even accounting for the bump.

Thank you for your suggestions to improve the Ca L-edge results. Unfortunately, the dip at ~ 347 eV (Supp Fig. 4) was already apparent in the raw data of some of the reference material (formate, citrate, calcite). Normalization to the incident flux did not improve this. We did not use these references quantitatively (i.e. linear combination fitting) so it has no impact on our results.

Replotting Fig S2 after adjusting our normalization parameters (see comment below) reduced post-edge baseline drift which likely improved the quality of the spectra.

We then use the normalized value to recalculate the L2/L3 splitting ratio (Fig S3). Expectedly, some of the values were different, however, as before, Ca-treatment did not seem to affect the splitting ratio.

SI methods:

Please include the normalization parameters in Athena, where you placed E0, the pre and post edge bounds, etc. for both the Ca and C edges on your bulk NEXAFS SI.

We have added the normalization parameters to the SI.

Cosmidis, J., Benzerara, K., Nassif, N., Tyliszczak, T., Bourdelle, F., 2015. Characterization of Ca-

phosphate biological materials by scanning transmission X-ray microscopy (STXM) at the Ca L_{2,3}-, P L_{2,3}- and C K-edges. *Acta Biomater* 12, 260-269.

Kalinichev, A.G., Kirkpatrick, R.J., 2007. Molecular dynamics simulation of cationic complexation with natural organic matter. *European Journal of Soil Science* 58(4), 909-917.